# Analytical realization of complex thermal meta-devices

Weichen Li[1], Ole Sigmund ®[2] & Xiaojia Shelly Zhang ®[1,3,4] ✉

Fourier's law dictates that heat flows from warm to cold. Nevertheless, devices can be tailored to cloak obstacles or even reverse the heat flow. Mathematical transformation yields closed-form equations for graded, highly anisotropic thermal metamaterial distributions needed for obtaining such functionalities. For simple geometries, devices can be realized by regular conductor distributions; however, for complex geometries, physical realizations have so far been challenging, and sub-optimal solutions have been obtained by expensive numerical approaches. Here we suggest a straightforward and highly efficient analytical de-homogenization approach that uses optimal multi-rank laminates to provide closed-form solutions for any imaginable thermal manipulation device. We create thermal cloaks, rotators, and concentrators in complex domains with close-to-optimal performance and esthetic elegance. The devices are fabricated using metal 3D printing, and their omnidirectional thermal functionalities are investigated numerically and validated experimentally. The analytical approach enables next-generation free-form thermal meta-devices with efficient synthesis, near-optimal performance, and concise patterns.

Metamaterials/meta-devices exhibit extreme properties not found in nature and enable exotic functionalities once conceivable only in fiction, such as optically cloaking an object from its environment[1–3]. Originating in theoretical electromagnetism[1], the idea of transformation-based omnidirectional field manipulation through strategic material distribution quickly gained enormous attention and spread to multiple disciplines. Most prominently, cloaking of physical fields[4,5] is theoretically investigated and experimentally reproduced in the discipline of electromagnetism[2–4,6–9], elasticity[10–12], acoustics[13–17], and thermotics[18–22]. In the latter, exotic functions such as heat cloaking[23–27], rotating[28], concentrating[29], camouflaging[30,31], illusion[32], and encrypting[33] have been reproduced experimentally. In addition to transformation-based theories, direct use of topology optimization[34–36] and data-driven methods for generating global structures[37–41] or local microstructures[42,43] produce the desired functionalities for specific targeted boundary conditions and applied heat gradient directions and are thus non-

omnidirectional. We note that this class of non-omnidirectional meta-devices does not employ transformation thermotics, and the meta-device's performance can significantly deteriorate when the direction of the applied heat is changed. The scope of this study is the omnidirectional class of thermal meta-device, where furthermore the conductivity distribution is obtained analytically and free of iterative procedures at a fraction of the cost of the above-mentioned numerical approaches.

Most fabricated meta-devices have simple and regular domains such as circles and ellipses, which allow for simple analytical solutions. For irregular domains favored by different applications, there exist no intuitive or analytical solutions. How to efficiently realize physically consistent, fabricable, and well-connected microstructures accurately, i.e., perform the de-homogenization that produces the spatially graded and highly anisotropic thermal conductivity for arbitrary geometries thus remains a major challenge[44]. The full resolution of the challenge is essential for overcoming the barrier between theory and

[1]Department of Civil and Environmental Engineering, University of Illinois Urbana-Champaign, 205 North Mathews Ave, Urbana, IL 61801, USA. [2]Department of Civil and Mechanical Engineering, Technical University of Denmark, Koppels Allé, Building 404, Kongens Lyngby 2800, Denmark. [3]Department of Mechanical Science and Engineering, University of Illinois Urbana-Champaign, 1206 W. Green St, Urbana, IL 61801, USA. [4]National Center for Supercomputing Applications, Urbana, USA. ✉e-mail: zhangxs@illinois.edu

realization and enabling the practical application of astounding thermal functionalities.

If not solvable by simple intuition, the state-of-the-art strategies to address the de-homogenization challenge is to divide the irregular domains into many square unit cells and use computational morphogenesis approaches (such as topology optimization) with numerical homogenization to inversely design the locally graded unit cell microstructures[26,44–46]. Although producing 3D-printable structures, the reliance on numerical design demands high computational cost and heavy post-processing to ensure inter-cell connection while producing overly complex and potentially sub-optimal structures. Replacement of numerical design with data-driven approaches can effectively reduce the cost[47], but the resulting structures remain sub-optimal and geometrically complex.

Contrary to the indication in[44,45] that the anisotropic and heterogeneous conductivities are difficult to realize by layered structures, the complete spectrum of a two-dimensional (2D) composite's conductivity can indeed be fully and analytically achieved by simple rank-2 laminates[48–52]. This suggests that the popular numerical inverse design approaches may have needlessly complicated the problem and produced sub-optimal performance. Furthermore, the conventional domain discretization into uniform-size quadrilateral unit cells is unnecessary in the current context and could restrict the microstructure patterns. Bypassing the state-of-the-art numerical design of local anisotropic properties, we recast our perception and revolutionize the practice of how thermal meta-devices can be realized and how they can be not only physically effective but also esthetically pleasing.

This study thus departs from the popular numerical route and develops a straightforward and highly efficient analytical de-homogenization framework to generate asymptotically consistent, high-performance, and fabricable thermal metamaterial structures with arbitrary shapes. The framework relies upon analytical rank-2 laminates with closed-form solutions of homogenized thermal conductivity and a de-homogenization technique that smoothly and seamlessly maps the laminates to any specified complex domains without severely perturbing their asymptotic properties. The computational cost of the realization procedure is negligible, and the resulting microstructures are concise, smooth, naturally well-connected, and readily 3D printable, and significantly outperform state-of-the-art approaches. The microstructures' elegant and intriguing patterns also enable direct perception of the underlying theoretical property distribution and anisotropy. We generate and numerically investigate a thermal cloak, a rotator, and a concentrator, and fabricate them using metal 3D printing. The realized omnidirectional thermal functionalities are experimentally reproduced and validated. The analytical and iteration-free framework with several developed techniques constitutes an efficient, robust, and distinctive paradigm for the realization of next-generation thermal meta-devices with arbitrary geometry, optimal performance, and elegant appearance.

## Results

This study focuses on steady-state heat conduction in a 2D medium without heat sources or sinks, which is mathematically described by $\nabla \cdot (\kappa \nabla T) = 0$ with $T$ being the temperature field and $\kappa$ the anisotropic $2 \times 2$ temperature-independent thermal conductivity tensor of the medium. Based on this setup, we present the overall idea of the study in Fig. 1A. As the first step, the theoretical distribution of $\kappa$ that produces different thermal functionalities is obtained through transformation thermotics as $\kappa = \frac{1}{\det J} J \kappa_0 J^T$ [20], where $J$ is the Jacobian matrix corresponding to the coordinate transformation from a reference coordinate system, and $\kappa_0$ is the thermal conductivity tensor of the base material in the reference coordinate system often chosen to be isotropic, i.e., $\kappa_0 = \kappa_0 I$ with $I$ being the identity matrix. Different coordinate transformations through $J$ produce different thermal meta-devices. Here, we focus on thermal cloaks, rotators, and concentrators.

Their corresponding coordinate transformation and the resulting $\kappa$ are given in Supplementary Note 1. The specific geometries of the meta-devices in this study are provided in Supplementary Note 2.

The obtained $\kappa$ generally features intense spatial variation and strong anisotropy, and its engineering realization requires composites that, when compounded with different constituent proportions, produce a wide range of anisotropic homogenized or effective thermal conductivities. Here, we adopt an analytical approach based on rank-2 laminates and the de-homogenization technique[53–57]. With their simple microstructures, rank-2 laminates have been shown to cover the entire range of physically achievable thermal conductivities for 2D composites[52], and therefore, our approach does not sacrifice any design freedom. As will be demonstrated, the proposed analytical approach can efficiently create high-resolution, well-connected, fabricable, and physically interpretable microstructures for arbitrary-shaped omnidirectional thermal meta-devices such as those in Fig. 1A. We hereafter refer to the more conductive constituent of the laminate as Material A and the less conductive as Material B.

The proposed de-homogenization technique is depicted in Fig. 1B. Given the $\kappa$ distribution from transformation thermotics, we first compute its eigenvalue ($\kappa_1$ and $\kappa_2$) and eigenvector ($v_1$ and $v_2$) fields. The eigenvalues are used for determining the volume fractions of Material A at the two scales of the rank-2 laminate ($f_1$ and $f_2$) through closed-form inverse homogenization, which is stated as finding $f_1$ and $f_2$ such that the resulting laminate's homogenized principal conductivities are equal to the eigenvalues (see Supplementary Notes 3.1 and 3.2 for details). The two-scale laminate is then analytically converted to a single-scale microstructure (see Supplementary Note 3.3 for the analytical conversion) defined by the widths ($w_1$ and $w_2$) of Material A in the two orthogonal directions with mild perturbation to the homogenized conductivity (see Supplementary Note 3.3 for error analysis).

The eigenvectors $v_1$ and $v_2$ are used for determining the orientations of the microstructure via a conformal mapping $\phi := [\phi_1, \phi_2] : \mathbb{R}^2 \to \mathbb{R}^2$ such that $\nabla \phi_i = p e_i$ with scalar $p > 0$ and $e_i \cdot v_i = 0$. The unit vector $e_i$, termed guiding vector herein, is aligned with $\nabla \phi_i$ and perpendicular to $v_i$. For the 2D setup, we take $e_1 = v_2$ and $e_2 = v_1$. The mapping $\phi_i$ preserves the shape and asymptotic properties of the microstructures and is obtained analytically for circular domains (see Supplementary Note 4 for the analytical solution). Only for complex domains, a numerical least-square solution is needed for finding the mapping $\phi_i$ and is obtained as follows[54] (take $\phi_1$ for illustration, same procedure for $\phi_2$).

$$\min_{\phi_1} \int_\Omega ||\nabla \phi_1 - e_1||^2 \, d\Omega \\ \text{s.t.} \quad \nabla \phi_1 \cdot e_2 = 0 \tag{1}$$

The least-square problem (1) is solved using a penalty approach based on the Finite Element Method (FEM, see Supplementary Note 5.1 for details) for the two directions. The computational cost for solving both $\phi_1$ and $\phi_2$ with a fine mesh on a desktop workstation is less than 30 s, which makes up the major cost of the total procedure.

Importantly, in (1), the domain $\Omega$ represents only the meta-device region excluding the core and is therefore non-simply connected. This would nullify the gradient representation of a wide range of $e_i$ fields, such as one circulating the core. To resolve this issue, we propose a special treatment that uses a zero-width gap to cut through the domain and additionally requires the $\phi_i$ on the two sides of the gap to differ by only an unknown constant. This treatment provides a simply connected domain while retaining identical $\nabla \phi_i$ on the two sides of the gap; the latter is necessary for the smooth structure transition at the gap. The proposed treatment enables the gradient representation of diverse $e_i$ fields while ensuring smooth and well-defined structures.

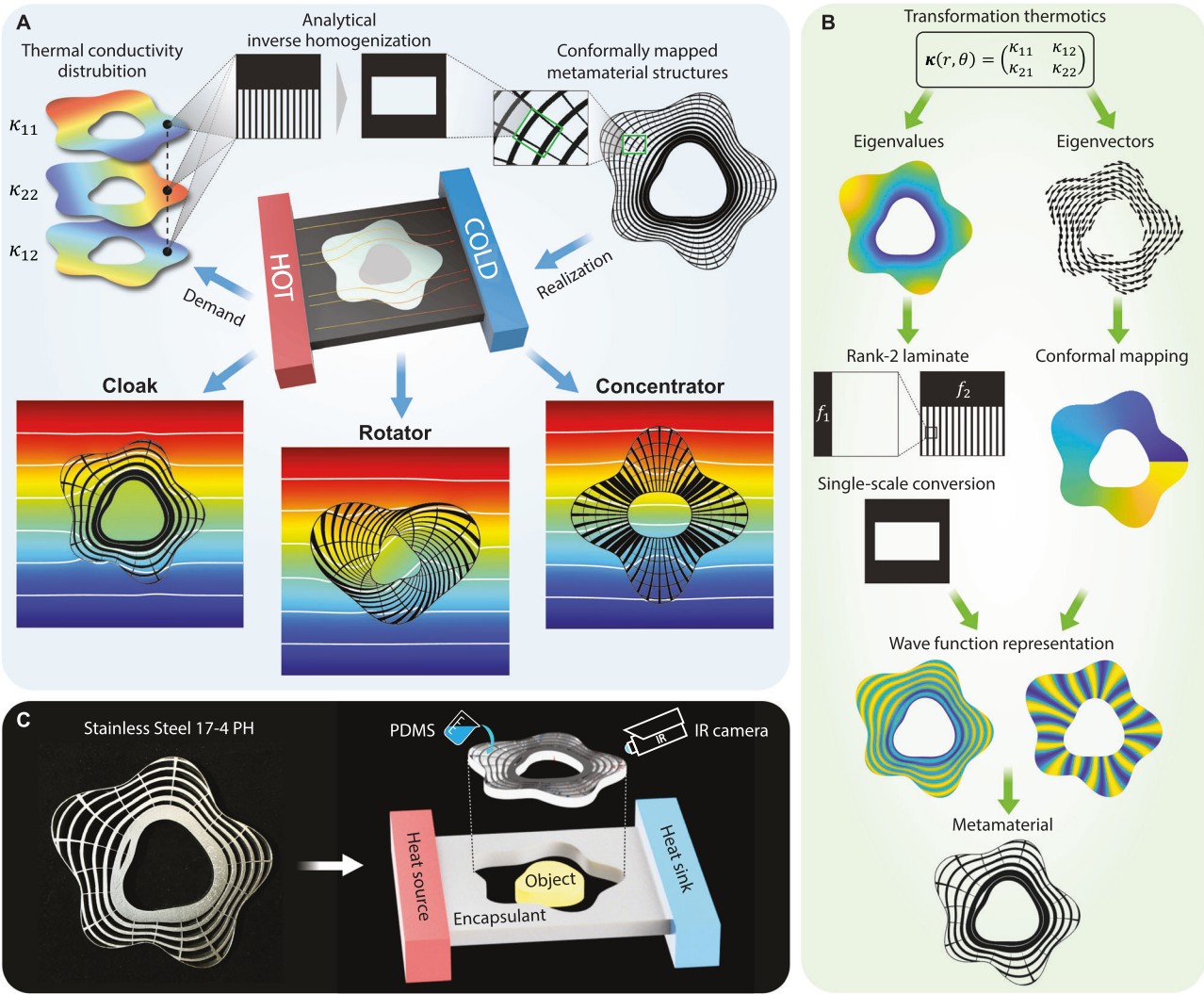

**Fig. 1 | Analytical freeform omnidirectional thermal meta-devices. A** Basic procedure of the analytical framework and its generated thermal cloak, rotator, and concentrator; **B** steps of the de-homogenization technique: from thermal conductivity field $\kappa$ to microstructures; **C** fabricated specimen and experimental setup.

Details of the special treatment are provided in Supplementary Note 5.2.

The inverse homogenization and the mapping $\phi_i$ from (1) determine the distribution of microstructural geometry and orientation for the whole domain. This information is represented through the following wave level set function[55].

$$\rho_i(r,\theta) = -\cos\left(\frac{2\pi}{\varepsilon_i}\phi_i(r,\theta)\right) + \cos(\pi w_i(r,\theta)), \; i = 1,2 \quad (2)$$

where $\varepsilon_i$ is the user-defined feature size parameter with larger values producing sparser but thicker members. Finally, the Material A part of the thermal metamaterial structure is defined by the set: $\{(r,\theta)|\rho_{min}(r,\theta) \le 0\}$ with $\rho_{min} := \min\{\rho_1,\rho_2\}$[55] while the rest of the domain is occupied by Material B. The structures of several meta-devices produced by the de-homogenization approach are shown in Fig. 1A.

Given any two constituents (Materials A & B), the achievable range of the homogenized $\kappa$ for rank-2 laminates can be analytically and explicitly obtained based on the corresponding rank-1 laminates' homogenized conductivity (see Supplementary Note 3.2 for details). This allows for instant evaluation of whether the two chosen constituents can achieve most of the target $\kappa$. In general, the achievable range of homogenized $\kappa$ can be made large by adopting a highly

conductive Material A and an insulating Material B. However, as will be shown later, overly conductive Material A can potentially result in very thin members in the final de-homogenized structures that may be challenging to fabricate. Hence, appropriate choices of the two constituents shall consider both the achievable range of homogenized properties and the fabricability of the final structures. This study uses steel 17-4PH alloy ($\kappa_{Steel} = 17.9\,\mathrm{Wm^{-1}K^{-1}}$) as Material A and Polydimethylsiloxane (PDMS, DOW SYLGARD™ 184, $\kappa_{PDMS} = 0.16\,\mathrm{Wm^{-1}K^{-1}}$) as Material B for the meta-device, and Thermal Conductive Encapsulant (DOW DOWSIL™ TC-6020, $\kappa_0 = 2.72\,\mathrm{Wm^{-1}K^{-1}}$) as the background material.

**Thermal cloak**

We first focus on the realization of a thermal cloak for a flower-shaped domain. The expressions of the inner and outer boundaries are provided in Supplementary Note 2. The fields of eigenvalues in the radial ($\kappa_1$) and circulating ($\kappa_2$) directions are shown in Fig. 2A. As the location moves from the inner to the outer boundaries, $\kappa_1$ increases from 0 to a moderate value, and $\kappa_2$ decays rapidly from high values. The corresponding distributions of volume fractions ($f_1$ and $f_2$) of the rank-2 laminate are analytically obtained and shown in Fig. 2A, showing a similar distribution as $\kappa_1$ and $\kappa_2$. The corresponding guiding vector fields ($\mathbf{e}_1$ and $\mathbf{e}_2$) are shown in Fig. 2B, representing a circulating and a diverging vector field, respectively. The resulting pseudo-conformal

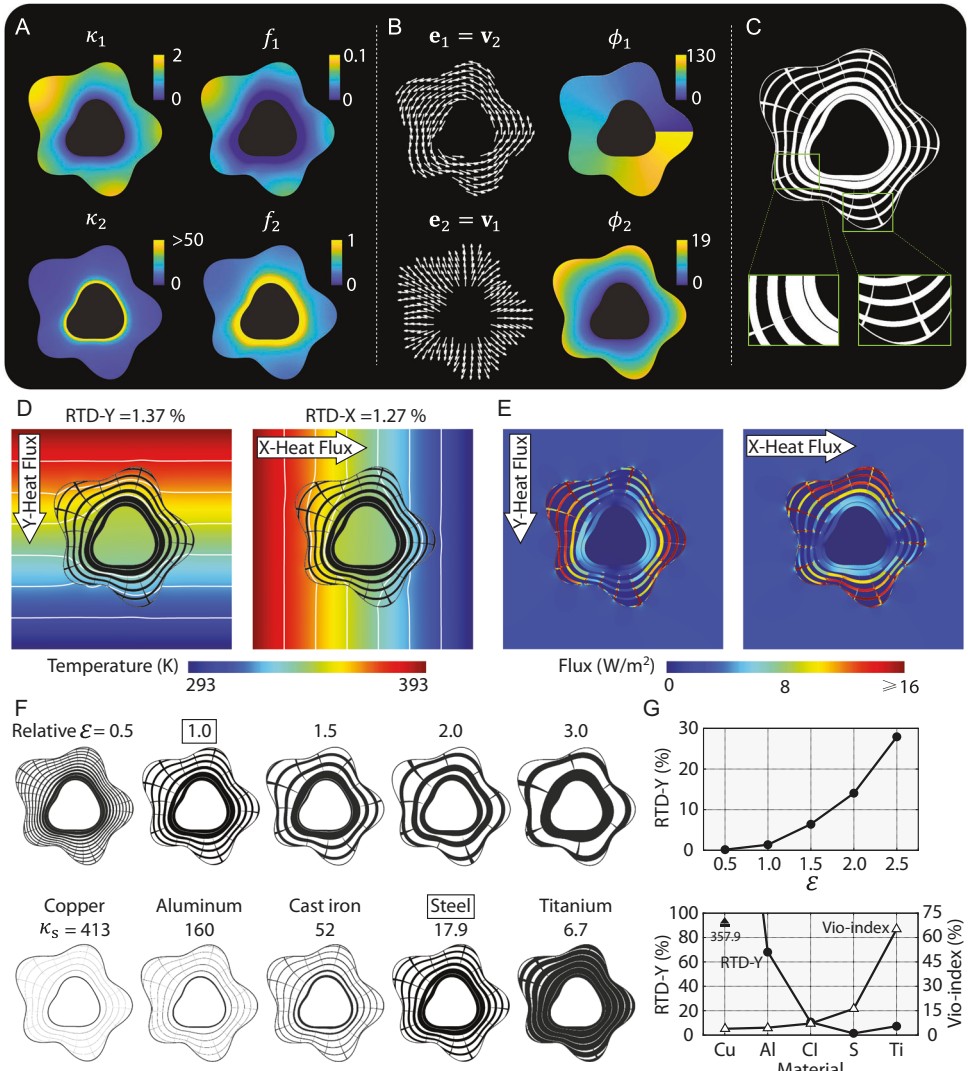

**Fig. 2 | Flower-shaped thermal cloak generated by the analytical framework.**
**A** Distributions of eigenvalues $\kappa_1$ and $\kappa_2$ of $\kappa$ and resulting Material A volume fractions $f_1$ and $f_2$ of rank-2 laminates; **B** distributions of guiding vectors $e_1$ and $e_2$ of $\kappa$ and resulting mappings $\phi_1$ and $\phi_2$; **C** de-homogenized structure of the thermal cloak; **D**, **E** temperature and heat flux distribution of the thermal cloak (FE simulation) under Y- and X-direction applied heat flux and the corresponding Relatve Temperature Difference (RTD) values; **F** thermal cloaks obtained with various feature size parameter and Material A; **G** RTD and Vio-index values of structures in (**F**).

mapping $\phi_1$ and $\phi_2$ obtained by solving (1) are shown in Fig. 2B. Note the discontinuity of $\phi_1$ at the right is the result of the special treatment to ensure simply connected domains (see Supplementary Note 5.2 for the special treatment).

The above information is input to the level set function representation (2) with feature size $\varepsilon_1 = 6.963$ and $\varepsilon_2 = 3.142$ (see Supplementary Note 5.2 for details about determining feature sizes) and generates the device in Fig. 2C. Detailed steps for generating the structure are provided in Supplementary Note 5.3. All members of the structure are naturally well-connected without any post-processing. The orientation and size distributions agree with the $\kappa$ distribution. The circulating and radial members intersect orthogonally by construction as they align with the guiding vector fields, and their sizes vary spatially and reflect the eigenvalue and volume fraction distributions. Near the inner boundary, the size of the circulating members is large while that of the radial members is close to zero. As the location moves away from the inner boundary to the outer, the sizes of the radial members increase while those of the circulating decrease. Note that the circulating member at the inner boundary conforms to the boundary's geometry while those near the outer boundary are non-conforming in general.

The cloaking performance of the device is post-evaluated and verified under heat conduction using the FEM (see Supplementary Note 6 for details), and the resulting temperature distributions and isotherms under vertically and horizontally applied heat flux are shown in Fig. 2D. For both cases, the isotherms outside the cloak remain straight and uniform, and the Relative Temperature Differences (RTD, see Supplementary Note 7 for definition, RTD-Y and RTD-X, respectively, correspond to Y- and X-direction applied heat) that evaluate the relative 2-norm temperature error compared to a homogeneous background medium are 1.37% and 1.27%, demonstrating very effective thermal cloaking and out-competing the state-of-the-art in[47]. The heat flux distributions are shown in Fig. 2E, showing a largely homogeneous value outside the cloak with some local fluctuations near the outer boundaries due to the finite length scale $\varepsilon_i$ of the realized microstructures. The heat flux in the PDMS core is close to zero, indicating that the cloak will hide any inclusion regardless of what material it is made from. Within the cloak, heat flux travels mainly

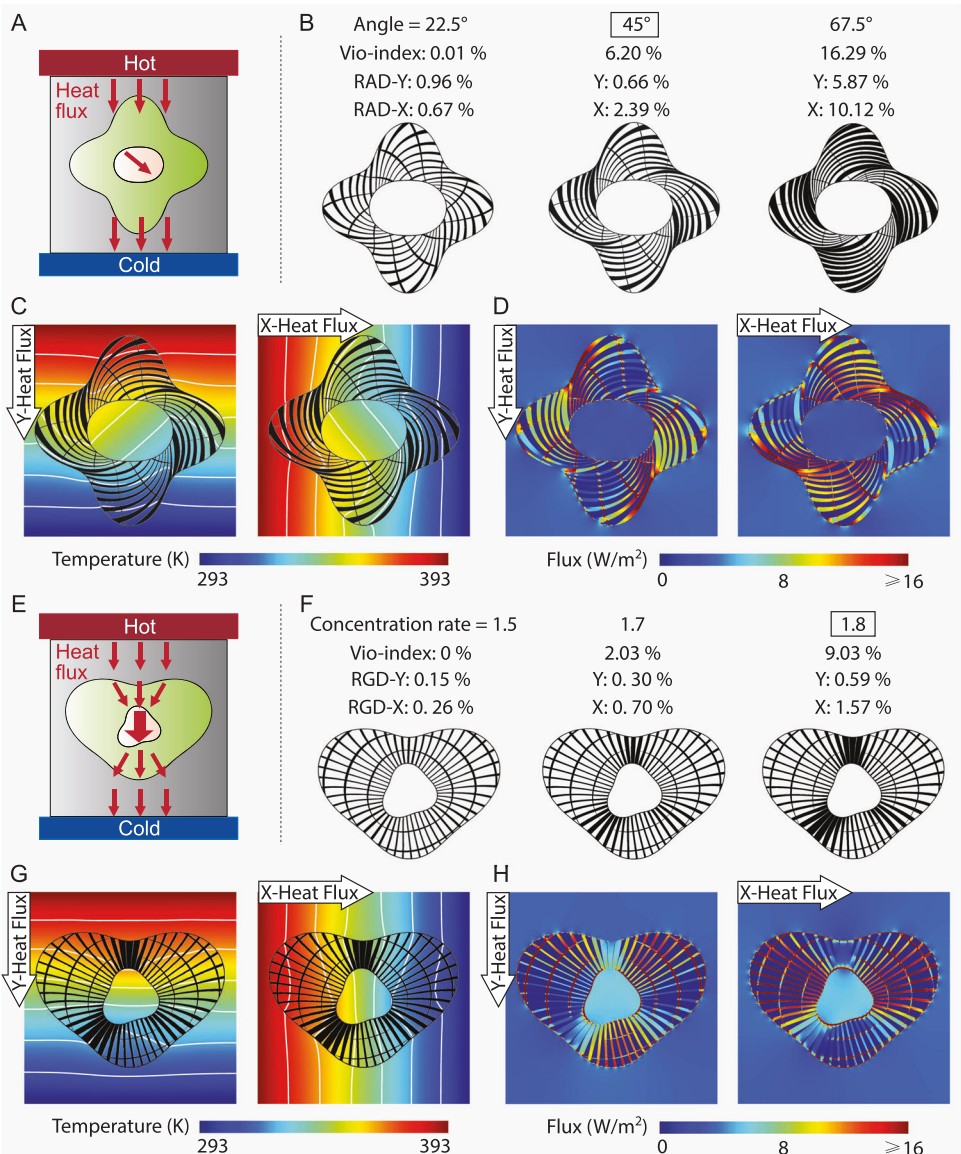

**Fig. 3 | Shuriken-shaped thermal rotator and heart-shaped thermal concentrator generated by the analytical framework. A** Illustration of thermal rotator: heat flux in the core is rotated by a specified angle without perturbing the background's heat flux distribution; **B** thermal rotators obtained with various specified rotation angles and their Vio-index values and Relative Angular Difference (RAD) values under Y- and X-direction applied heat (RAD-Y and RAD-X); **C, D** temperature and heat flux distributions of the 45-degree thermal rotator (FE simulation) under Y- and X-direction applied heat flux; **E** Illustration of thermal concentrator: heat flux in the core is magnified by a specified concentration level without perturbing the background's heat flux distribution; **F** thermal concentrators obtained with various specified concentration levels and their Vio-index values and Relative Gradient Difference (RGD) values under Y- and X-direction applied heat (RGD-Y and RGD-X); **G, H** temperature and heat flux distributions of the 1.8-concentration level thermal concentrator (FE simulation) under Y- and X-direction applied heat flux.

through the steel members as expected, with higher flux in members along the applied temperature gradient.

The de-homogenized structure and its cloaking performance are influenced by the feature size $\varepsilon_i$ as demonstrated in the top row of Fig. 2F. The first row of structures in Fig. 2F are generated with various feature size parameters $\varepsilon_i$ (relative to the $\varepsilon_i$ of the structure in Fig. 2C), and their RTD values are shown in the circular markers in Fig. 2G. In general, a decrease in feature size improves the performance for its more accurate approximation of the $\kappa$ distribution, but small feature sizes are harder to fabricate. Structures with moderate feature sizes such as the one with 1.5 relative feature size are more fabricable while retaining a relatively low RTD value.

Change of Material A given the background and Material B also significantly alters the performance as demonstrated by the bottom row of structures in Fig. 2F and their RTD values in Fig. 2G. More conductive Material A generally expands the achievable range of homogenized conductivity as evidenced by the Violation Index (Vio-index, defined as the ratio of area with unachievable target $\kappa$ over the total meta-device area) in Fig. 2G. While larger achievable ranges are theoretically more advantageous, they can result in extremely thin members (thinner than the finite element size) that in turn deteriorate the performance evaluation and fabricability, as shown in the growing RTD-Y values in Fig. 2G as Material A's conductivity grows. On the other hand, Material A with overly low conductivity (such as Titanium) will have a high Vio-index that will also worsen the performance because a large proportion of target $\kappa$ cannot be accurately achieved, although the resulting member size is large and fabrication-friendly. Hence, a good performance requires a balance between the range of achievable

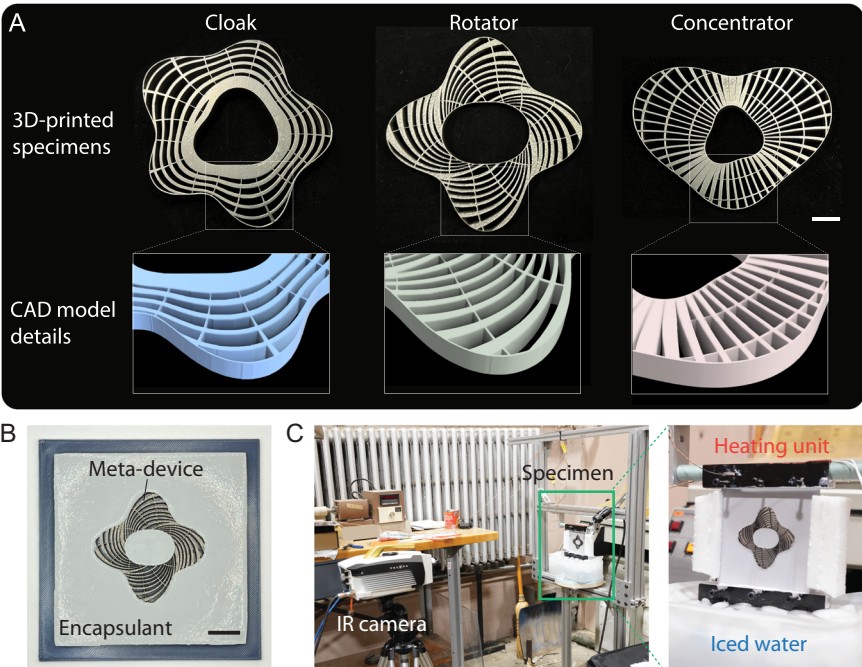

**Fig. 4 | Fabricated specimens and experimental setup. A** Metal 3D-printed Material A part of the thermal cloak, rotator, and concentrator (scale bar: 10 mm) and their CAD models; **B** fabricated thermal device with background material (scale bar: 20 mm); **C** experimental setup for acquiring temperature distribution and applying constant temperatures on the top and bottom ends of the specimen.

homogenized properties and the feature size, both directly depending on the selection of the constituents. In the proposed method, we can analytically state what material is needed to realize a specific structure with the best performance and fabricability given a background and Material A. In the current setup, our choice of Steel and PDMS produce the best RTD values, even though the Vio-index is rather high (20%). This also demonstrates the robustness of the method as it can produce satisfactory thermal functionalities even if a proportion of the target $\kappa$ is unachievable locally by the chosen constituents. The robustness is also manifested in the created cloak's insensitivity to core (cloaked) materials (see Supplementary Note 8.1 for extended investigations).

## Thermal rotator and concentrator

We now use the proposed analytical framework to generate thermal rotators with a complex domain as illustrated in Fig. 3A, where the heat flux in the core is rotated by a specified angle, and the heat flux in the background remains undisturbed. Here, we rotate the heat flux counter-clockwise by angles of 22.5°, 45°, and 67.5° and produce the structures shown in Fig. 3B. The devices consist of two families of orthogonal members with opposite spiral patterns and dissimilar sizes. When viewed from the outer boundary toward the core, the counter-clockwise members are much thicker in general than the clockwise members as they align with the specified rotation. As shown by the Vio-index of the three devices in Fig. 3B, raising the specified rotation angle yields more extreme and anisotropic target $\kappa$ that are more difficult to achieve with the two constituents, which in turn deteriorate the rotation performance measured by the introduced Relative Angular Differences (RAD) (see Supplementary Note 7 for definition), which quantifies the relative 2-norm error between the temperature gradient direction in the core and the specified rotation angle. The more extreme $\kappa$ also leads to larger size, density, and lengths of the counter-clockwise members while decreasing the size of the clockwise members. The member orientations and sizes reflect the eigenvectors and eigenvalues of a thermal rotator's $\kappa$ distributions, and all members are naturally well connected and smoothly varying in space.

The performance of the 45° structure is post-evaluated using FEM under vertically and horizontally applied heat flux as shown in Fig. 3C. The isotherms demonstrate the rotated temperature gradient (and hence heat flux) in the core and the constant temperature gradient in the background for both cases. The corresponding heat flux distributions are shown in Fig. 3D, demonstrating a uniform heat magnitude in the background and in the core. The RAD of the three devices under Y- and X-direction applied heat flux (RTD-Y and RTD-X) are given in Fig. 3B. In general, a larger rotation angle requires more extreme $\kappa$ (analytically given) that could be harder to achieve accurately using the given constituents, leading to worsened performance. For the 22.5° and 45° structures, the RADs are low, indicating the specified heat flux rotation in the core is accurately achieved. Precise realization of larger rotation angles requires Material A with higher conductivity (such as copper).

We now focus on generating thermal concentrators using a heart-shaped domain as illustrated in Fig. 3E. Under an applied temperature gradient, the concentrator would magnify the temperature gradient (and hence heat flux) in the homogeneous core by a specified concentration rate. The expression of the $\kappa$ is given in Supplementary Note 1. Figure 3F shows three de-homogenized concentrators with concentration rates of 1.5, 1.7, and 1.8, respectively. Like the cloak, the concentrator also consists of radial and circulating members with spatially varying sizes, but the radial ones are generally larger than the circulating ones as they facilitate the heat flow into the core, realizing the concentration. Higher concentration rates require more extreme and anisotropic $\kappa$ values as manifested by the larger radial members and smaller circulating members. As a side note, uni-directional concentration have been solved and require only rank-1 laminates for their realization[58]. Because the more anisotropic targets are harder to achieve as shown by the growing Vio-index values, they worsen the overall performance measured by the introduced Relative Gradient Difference (RGD, see Supplementary Note 7 for definition), which is the relative 2-norm error between the concentrated temperature gradient and the ideal specified value. Nevertheless, the RGD values of the three

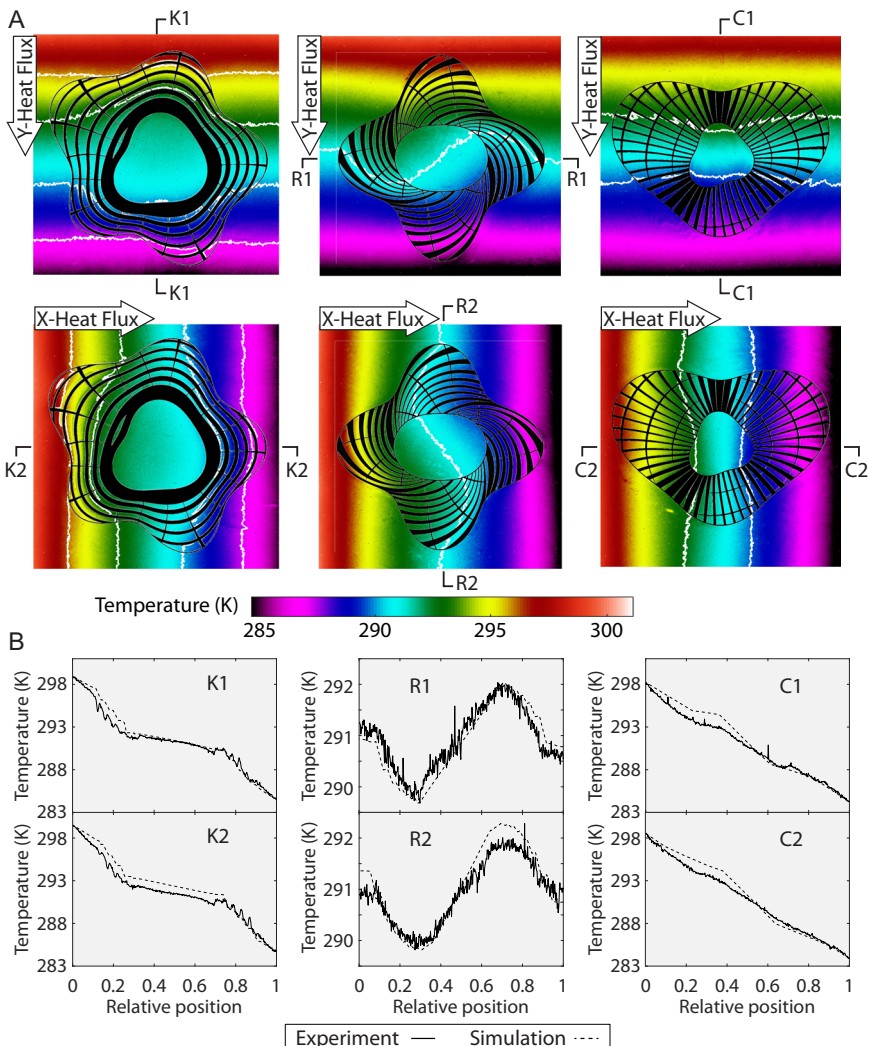

Temperature (K)
285   290   295   300

**Experiment** ——   **Simulation** ·····

**Fig. 5 | Experimental results of thermal cloak, rotator, and concentrator under Y-direction and X-direction applied heat flux. A** (from left to right) experimental temperature distributions of the thermal cloak, rotator, and concentrator; **B** comparison between simulation and experimental temperature profiles of the middle sections.

structures are generally small, showing an accurate focus of heat flux to the specified level irrespective of the local achievability of the target $\kappa$.

The FEA evaluation of the 1.8-concentration rate meta-device is shown in Fig. 3G, demonstrating the magnified temperature gradient in the core compared to the background. The concentration can be more clearly observed in the fringe plot of heat flux in Fig. 3H, where the value in the core is considerably higher than the background's and is relatively uniform apart from some fluctuations near the interface.

Extended investigations on the rotator's and concentrator's performance with various feature sizes are provided in Supplementary Note 8.2. In general, the rotator's performance is more sensitive to feature size than the concentrator as the RAD value deteriorates faster as the feature size increases.

**Experimental validation**

We fabricate and validate the performance of the cloak, rotator, and concentrator. The steel (Material A) parts of the meta-devices are 3D-printed using direct metal laser sintering (Proto Labs, Inc.) as shown in Fig. 4A, and the voids (also the core of the cloak) are filled with PDMS. The meta-device is then cast with the 120 mm × 120 mm background encapsulant to form the final device as demonstrated in Fig. 4B. The

cores of the rotator and concentrator are also cast with the encapsulant. The thickness of the meta-device and background is 4.5 mm. More fabrication details are provided in Supplementary Note 9. To apply the temperature difference in the experiment, we placed the material vertically with the top and bottom ends clamped by a metal plate with a slot. The top plate is electrically heated to 310 K (37 °C), and the bottom plate is soaked in a tank of iced water, as shown in 4C. Thermal grease is applied to the gaps between the metal plates and the material for smooth and uniform heat paths. A Telops FAST M100hd infrared camera is used to record the temperature distribution. The emissivity is calibrated based on a reference material. To better represent the adiabatic boundary conditions on the left and right boundaries, we clamp a foam bar on each side and cover the ice bath with a lid to prevent convection downdraft. We report the experimental result after the temperature reaches the steady state (see Supplementary Note 9 for more details about the experimental setup).

The experimental temperature distributions of the thermal cloak, rotator, and concentrator under X-direction and Y-direction applied heat flux are shown in Fig. 5A. The targeted omnidirectional thermal functionalities are physically reproduced. For the cloak, the temperature gradient in the background is uniform and largely unperturbed while the temperature gradient inside the core is very

small as demonstrated by the isotherms. As the core is made of PDMS which differs considerably in thermal conductivity from the background material, the experiment result demonstrates an effective cloaking of an object inside a medium. For the rotator, the temperature gradient inside the core is rotated as clearly seen via the isotherm, and the background temperature gradient remains largely uniform. For the concentrator, the heat concentration effect is demonstrated via the two isotherms whose distance inside the core is much smaller than the outside. For the non-steady-state responses of the three devices during the experiment, the readers are referred to Supplementary Movies 1–3.

For a more detailed investigation, we plot the temperature profiles at the middle sections (K1 and K2 for cloak, R1, and R2 for rotator, and C1 and C2 for concentrator as indicated in Fig. 5A) against their numerical counterparts in Fig. 5B. In these simulations, the hot-end and cold-end temperatures are set to the same as in the experiment. The experimental profiles generally match the simulation well in both applied-heat directions, again demonstrating the robustness and reliability of the method. For the cloak, the plateau represents the nearly constant temperature inside the core, and for the rotator, the gradient of the profile reflects the rotated temperature gradient. For the concentrator, the temperature gradient inside the core is slightly larger than the outside.

## Discussion

Omnidirectional heat manipulation was thus far achieved for simple domains but immature for general geometries due to the lack of an asymptotically consistent framework to efficiently generate fabricable devices with optimal performance. This study closes the gap by developing an analytical framework based on rank-2 laminates and de-homogenization techniques, leading to a new paradigm for producing freeform thermal meta-devices that reflect analytical material property distributions. Significantly departing from existing approaches, the analytical framework generates highly efficient meta-devices with high-resolution, well-connected, and esthetically illuminating micro-structures at negligible computational cost. The effectiveness and near-optimal performance of the created meta-devices are demonstrated through the thermal cloak, rotator, and concentrator, and the desired functionalities are validated experimentally.

## Methods
### Transformation thermotics

This study uses transformation thermotics to obtain the analytical distributions of thermal conductivity $\kappa$. It transforms a reference polar coordinate into a physical polar coordinate. Different transformation rules yield different meta-devices. Details of the transformation procedure are provided in Supplementary Notes 1. The domains of the meta-devices are prescribed by defining the inner and outer boundaries. Details of the definitions are provided in Supplementary Notes 2.

### Inverse homogenization of Rank-2 laminates

The inverse homogenization, i.e., find the Material A volume fractions $f_1$ and $f_2$ given the two conductivity eigenvalues $\kappa_1$ and $\kappa_2$, is obtained analytically. The obtained rank-2 laminate defined by $f_1$ and $f_2$ is then analytically and approximately converted into a single-scale micro-structure based on a volume-equivalent condition. Details of the inverse homogenization and conversion are provided in Supplementary Note 3.

### (Pseudo-)conformal mapping

The (pseudo-)conformal mapping is used to map the local rank-2 laminates into a smooth and asymptotically consistent global meta-material. For the case of circular cloak and circular concentrator, the mapping can be obtained in closed form and is strictly conformal (angle-preserving). The procedure for the analytical solution is

provided in Supplementary Note 4. For complex domains, closed-form solutions are not attainable in general, and we use numerical least-square solutions (Eq. (1)) based on FEM. The result is a pseudo-conformal mapping that preserves angles in the principal directions. Details of the numerical solution and subsequent generation of the metamaterial are provided in Supplementary Note 5.

### Performance measures

The FE post-evaluation of the meta-devices' thermal performance is elaborated in Supplementary Note 6. The quantitative definition of the performance measures (RTD, RAD, and RGD) for different meta-devices is provided in Supplementary Note 7.

### Parametric studies

The study carries out comprehensive parametric studies of the meta-devices based on FEM. In addition to those presented in the main text, the study also investigates the influence of core material on the performance of thermal cloak and the influence of feature size on the performance of thermal rotator and concentrator. Detailed discussions of these investigations are presented in Supplementary Note 8.

### Fabrication and experimental setup

The 3D-printed metal part is cast with PDMS and cured to form a one-piece structure. The structure is then placed in the middle of a square plastic frame and cast with the encapsulant (DOW DOWSIL™ TC-6020) to form the final specimen. For the experiment, one side of the specimen is inserted into a bar-shaped electric heater with temperature control, and the opposite side is soaked in a tank of iced water. This creates the temperature gradient in the specimen, which is captured by the infrared camera. More details of the fabrication and experiment are provided in Supplementary Note 9.

## Data availability

The data generated in this study are available from the main text or Supplementary Information. The design of the meta-devices and their numerical simulation data have been deposited in Zenodo with https://doi.org/10.5281/zenodo.11211414.

## Code availability

The codes to generate the results are available from the corresponding author upon request.

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

## Acknowledgements

We want to thank Peter Dørffler Ladegaard Jensen for sharing his implementation on parts of the de-homogenization technique. The

experiment in this study was carried out in part in the Advanced Materials Testing and Evaluation Laboratory, Materials Research Laboratory, University of Illinois. This material is based upon work supported by the Air Force Office of Scientific Research under award number FA9550-23-1-0297. In addition, author X.S.Z. acknowledges the support from U.S. National Science Foundation (NSF) CAREER Award CMMI-2047692 and NSF Award CMMI-2245251. Author O.S. acknowledges the support from the Villum Foundation Villum Investigator Project "InnoTop", Denmark.

## Author contributions

O.S. and X.S.Z. conceived the research and supervised the research project. W.L., O.S., and X.S.Z. developed the methodology, carried out the theoretical and numerical investigation, and drafted and revised the manuscript. W.L. performed the experiment. X.S.Z. secured the funding support and provided the resources.

## Competing interests

The authors declare no competing interests.
