## [Peer Review File · Nature Communications]

Analytical realization of complex thermal meta-devicesReviewers' comments:

Reviewer #1 (Remarks to the Author):

In this manuscript, the authors adopted an analytical approach to de-homogenize the theoretical distributed thermal conductivity obtained through transformation thermotics into the discrete equivalent one. The simulation and experimental results validate well. However, there are some important issues that require clarification:

1. In Figure 1A, the color of "Thermal conductivity distribution" is confused. According to transformation thermotics, the thermal conductivity is anisotropic. So what does the color mean?
2. The process from "Thermal conductivity distribution" to "Analytical inverse homogenization" (Figure 1A) has not been described clearly in the manuscript. The thermal conductivity tensor relationship between these two parts is not clearly explained.
3. In Figure 1B, the variate of the equation in "Transformation thermotics" part should be r and θ .
4. In the third paragraph of "2 Results", the eigenvalues are used for determining the volume fractions of Material 1. But SI 3.1 and 3.2 do not mention the eigenvalues. This part is vague. Please explain it in detail.
5. It is still not clear how the final structure is designed with specific feature size and the other information that is input to the wave function. The authors should explain in detail how to get the final structure with pseudo-conformal mapping and feature size. Please take thermal cloak as an example to give a specific derivation.
6. In figure 5A, why there is no isotherm in thermal cloak but the other two include it?
7. There are some relevant papers that should be cited and discussed, such as Cell Reports Physical Science, 4: 101540, 2023; Advanced Materials, 30, 1707237(2018); Advanced Materials, 31, 1807849(2019); Materials Today, 45:120-141, 2021; International Journal of Heat and Mass Transfer, 176:121417, 2021.

Reviewer #2 (Remarks to the Author):

In this paper, microstructures are designed by evaluation using finite element analysis and optimization using Newton's method to achieve the thermal properties required by transformation thermotics for realizing thermal cloak and related metadevices. Although it is interesting overall for this reviewer, it is difficult to recommend that the presented study is innovative enough to be published in Nature Communications. In addition, it is very difficult to understand the position of this research in optimization research on thermal cloak (and its applications), and the paper simply claims to be the state of the art. The reasons why this paper is NOT sufficient for publication in Nature Communications are described below.

IMPACT on METHOD: Despite many papers have been published on thermal cloaking (and its applications) designed by numerical approaches including topology optimization, but there are almost no descriptions of them. The authors should comprehensively cite and organize previous studies using the density method, level set method, and homogenization method (including papers using dehomogenization), and explain the location and advantages of this research. However, no descriptions and paragraph on the previous works are written in the introduction.

Also, this study is very similar to the previous study [24], and the introduction need to include how this study is innovative compared to the study ref. [24] that used topology optimization using the density method for microstructure design. Of course, there are some differences from the references. However, when asked whether this has enough impact to be published in Nature Communications, this reviewer thinks it is NOT.

PERFORMANCE: Despite the very high degree of design freedom by designing the microstructure, the RTD of the designed structure is higher (worsen) than the equivalent value in some previous studies and it has not been improved.

QUALITY: (??) remains just before the SI equation (15) and this shows the quality of the manuscript.

Reviewer #3 (Remarks to the Author):

The so called "transformation physics" leads to designs, using composites at the microscale, that lead to sometimes surprising conclusions about what is possible. However, the designs are typically far from optimal and sometimes utilize constituent materials with extreme properties. By contrast the authors show how one can use relatively simple designs, incorporating rank 2 laminates at the microscale, that have a continuous morphing of structure on the mesoscale, and they verify their near optimal performance experimentally. This is a major advance and, although currently limited to two-dimensions, there is no reason that the work cannot be extended to three dimensions. However, a major flaw in the paper is that it perpetuates misconceptions about original discoveries. Notably, Pendry, Schurig and Smith were not the first to discover "transformation electromagnetism: this dates back to Dolin ("To the possibility of comparison of three-dimensional electromagnetic systems with nonuniform anisotropic filling", *Izvestiya Vysshikh Uchebnykh Zavedeni\u{u}\{i\}* Radiofizika" 4, 964--967 (1961)) who showed how to construct invisible inclusions via this method. A translation of Dolin's paper can be found on the internet. Nor were Pendry, Schurig and Smith and Leonhardt the first to discover cloaking. Transformation conductivity (including thermal conductivity), was used by Greenleaf, Lassas, and Uhlmann in 2003 to cloak conducting objects: (<http://iopscience.iop.org/0967-3334/24/2/353> and <http://dx.doi.org/10.4310/MRL.2003.v10.n5.a11>) In fact the results of Pendry, Schurig, and Smith are a simple combination of these ideas of Dolin and Greenleaf, Lassas, and Uhlmann. For acoustic cloaking it would be important to reference the work of Norris (<http://dx.doi.org/10.1098/rspa.2008.0076>).

On the second page it is stated that "the complete spectrum of composite's conductivity can indeed be fully and analytically achieved by simple rank-2 laminates". Nowhere here is started the all important fact that this is limited to two-dimensional composites (and that rank-3 laminates are needed in three dimensions). The references here should include those of Tartar and Murat (which can be found in books on the theory of composites). Otherwise it is a bit unfair, as Professor Kohn, on a trip to the Soviet Union, had communicated the results of Tartar and Murat to Lurie and Cherkaev who incorporated them in their work, without attribution (they claim they had proved the bounds in an alternative way but utilized the approach of Murat and Tartar as it was simpler). Lurie and Cherkaev had independently obtained the two dimensional bounds.

The authors talk about cloaking and concentrating fields between a hot plate and a cold plate. For the case where the hot and cold plates are fixed, this was solved using rank one laminates by Gibiansky, Lurie, and Cherkaev and Gibiansky in the paper *Zhurnal tekhnicheskoi fiziki* 58 (1988) 67-74, english translation in *Sov. Phys. Tech. Phys.* 33:38-42 (1988).

This can be made more clear if one considers the periodic extension of their solution. So that work needs to be cited.

A major deficiency is that I do not see the equations of thermal conduction used by the authors stated anywhere. Presumably the authors are not incorporating Joule heating which is a non-linear term, but without some statement of the equations it is hard to know what they are including.

Response to the reviewers

The authors are most grateful for the reviewers' insightful comments and suggestions. We have thoroughly revised the manuscript according to all the comments. The textual changes are highlighted in red color in the revised manuscript and the added citations are highlighted yellow. Below we address each comment in detail.

Reviewer #1 (Remarks to the Author):

In this manuscript, the authors adopted an analytical approach to de-homogenize the theoretical distributed thermal conductivity obtained through transformation thermotics into the discrete equivalent one. The simulation and experimental results validate well. However, there are some important issues that require clarification:

We thank the reviewer for the compliments and comments about our work. Below we address each comment in detail.

1. In Figure 1A, the color of “Thermal conductivity distribution” is confused. According to transformation thermotics, the thermal conductivity is anisotropic. So what does the color mean?

We thank the reviewer for pointing this out. We have modified Figure 1A to reflect the anisotropy by illustrating the distribution of the conductivity tensor components as follows.

2. The process from “Thermal conductivity distribution” to “Analytical inverse homogenization” (Figure 1A) has not been described clearly in the manuscript. The thermal conductivity tensor relationship between these two parts is not clearly explained.

We thank the reviewer for pointing this out. We have improved the description as follows.

“The eigenvalues are used for determining the volume fractions of Material A at the two scales of the rank-2 laminate (f_1 and f_2) through closed-form inverse homogenization, which is stated as finding f_1 and f_2 such that the resulting laminate's homogenized principal conductivities are equal to the eigenvalues (see SI 3.1 and SI 3.2 for details).”

3. In Figure 1B, the variate of the equation in “Transformation thermotics” part should be r and θ .

We thank the reviewer for the suggestion. We have modified Figure 1B as follows.

4. In the third paragraph of “2 Results”, the eigenvalues are used for determining the volume fractions of Material 1. But SI 3.1 and 3.2 do not mention the eigenvalues. This part is vague. Please explain it in detail.

We thank the reviewer for pointing this out. To clarify, we have comprehensively revised SI 3.2 to relate the eigenvalues to the rank-2 laminate as follows.

“Realizing the thermal meta-device requires the inverse homogenization of rank-2 laminates, that is, given the two eigenvalues of κ obtained from transformation thermotics, what are the f_1 and f_2 values such that the corresponding homogenized $\kappa_{R2,X}$ and $\kappa_{R2,Y}$ from (9) are equal to the two eigenvalues? The inverse problem has a closed-form solution attained by solving (9) and (10) for f_1 and f_2 given $\kappa_{R2,X}$ and $\kappa_{R2,Y}$ ”

$$f_1 = \frac{\kappa_A(\kappa_A\kappa_B - \kappa_A\kappa_{R2,X} - \kappa_B\kappa_{R2,X} + \kappa_{R2,X}\kappa_{R2,Y})}{-\kappa_A^3 + \kappa_A^2\kappa_B + \kappa_A\kappa_{R2,X}\kappa_{R2,Y} - \kappa_B\kappa_{R2,X}\kappa_{R2,Y}}$$

$$f_2 = \frac{\kappa_A(\kappa_A\kappa_B - \kappa_A\kappa_{R2,Y} - \kappa_B\kappa_{R2,Y} + \kappa_{R2,X}\kappa_{R2,Y})}{-\kappa_A^3 + \kappa_A^2\kappa_{R2,X} + \kappa_A\kappa_B\kappa_{R2,X} - \kappa_B\kappa_{R2,X}\kappa_{R2,Y}}$$

where physical values are $f_1, f_2 \in [0,1]$. In (11), $\kappa_{R2,X}$ and $\kappa_{R2,Y}$ are set equal to the two eigenvalues of κ .”

5. It is still not clear how the final structure is designed with specific feature size and the other

information that is input to the wave function. The authors should explain in detail how to get the final structure with pseudo-conformal mapping and feature size. Please take thermal cloak as an example to give a specific derivation.

We thank the reviewer for the suggestion. To comprehensively illustrate the process of generating the structures, we have added a section SI 5.3 and a figure (Figure 5) to the Supporting Information as follows.

“After obtaining the local microstructural parameters w_1 and w_2 (in 3.3) and the mapping ϕ_1 and ϕ_2 (in 5.1 and 5.2) as four scalar fields on the meta-device domain, we can generate the corresponding global structure through a wave function representation [5]. Specifically, we need two wave functions to represent the two groups of orthogonal laminates in the global structure, respectively. The wave functions are

$$\rho_i(r, \theta) = -\cos\left(\frac{2\pi}{\varepsilon_i}\phi_i(r, \theta)\right) + \cos(\pi w_i(r, \theta)), \quad i = 1, 2 \quad (20)$$

where the feature size parameter ε_i is user-defined and needs to satisfy the mild requirement discussed in 5.2.

To illustrate the procedure, we use the cloak as an example as shown in Figure 5. Given the four (discrete) fields of w_1 , w_2 , ϕ_1 , and ϕ_2 , we obtain the wave functions ρ_1 and ρ_2 using (20) with values plotted in the third column of Figure 5. Based on [5], the Material 1 (steel) parts for the first and second laminates are defined by the regions $\rho_1 \leq 0$ and $\rho_2 \leq 0$, respectively, which are shown in the fourth column of Figure 5. Note that relative widths and orientations of the laminates are consistent with the distributions of w_i and ϕ_i by construction. Finally, the metamaterial structure is obtained as the union of the two laminates as shown in the right-most plot of Figure 5, and the union set can be expressed as $\rho_{min} \leq 0$ with $\rho_{min} := \min\{\rho_1, \rho_2\}$. We want to emphasize that the process of generating the structure, i.e., computing the wave functions and taking the union, requires negligible computational cost and is free of post-processing for structural connectivity, which is a major advantage over state-of-the-art computation-based approaches.”

Figure 5: Process to generate the meta-device (cloak) after obtaining w_1 , w_2 , ϕ_1 , and ϕ_2 .

6. In figure 5A, why there is no isotherm in thermal cloak but the other two include it?

We thank the reviewer for pointing this out. We have added the isotherms to Figure 5A as follows.

We have also modified the description about the cloak experiment as follows.

“For the cloak, the temperature gradient in the background is uniform and largely unperturbed while the temperature gradient inside the core is very small as demonstrated by the isotherms.”

7. There are some relevant papers that should be cited and discussed, such as Cell Reports Physical Science, 4: 101540, 2023; Advanced Materials, 30, 1707237(2018); Advanced Materials, 31, 1807849(2019); Materials Today, 45:120-141, 2021; International Journal of Heat and Mass Transfer, 176:121417, 2021.

We thank the reviewers for suggesting these relevant studies. They have been cited in the first two paragraphs of the revised Introduction as follows.

“Metamaterials exhibit extreme properties not found in nature and enable exotic functionalities once conceivable only in fiction, such as optically cloaking an object from its environment [1, 2, 3]. Originating in theoretical electromagnetism [1], the idea of transformation-based omnidirectional field manipulation through strategically designed material distribution quickly gained enormous attention and spread to multiple disciplines. Most prominently, cloaking of physical fields [4, 5] are theoretically investigated and experimentally reproduced in the discipline of electromagnetism [4, 2, 3, 6, 7, 8, 9] elasticity [10, 11, 12], acoustics [13, 14, 15, 16, 17], and thermotics [18, 19, 20, 21, 22]. In the latter, exotic functions such as heat cloaking [23, 24, 25, 26, 27], rotating [28], concentrating [29], camouflaging [30, 31], illusion [32], and encrypting [33] have been designed and reproduced experimentally. In addition to transformation-based theories, direct use of topology optimization [34, 35, 36] and data-driven methods for designing global structures [37, 38, 39, 40] or local microstructures [41] can produce the desired functionalities for specific targeted load cases and directions and are thus non-omnidirectional. We note that this class of non-omnidirectional metamaterials does not employ transformation thermotics, and the meta-device’s performance significantly deteriorates when the direction of the applied heat is changed. The scope of this study is the

omnidirectional class of thermal metamaterial, where furthermore the conductivity distribution is obtained analytically and free of iterative procedures at a fraction of the cost of above-mentioned numerical approaches.”

Reviewer #2 (Remarks to the Author):

In this paper, microstructures are designed by evaluation using finite element analysis and optimization using Newton's method to achieve the thermal properties required by transformation thermotics for realizing thermal cloak and related metadevices. Although it is interesting overall for this reviewer, it is difficult to recommend that the presented study is innovative enough to be published in Nature Communications. In addition, it is very difficult to understand the position of this research in optimization research on thermal cloak (and its applications), and the paper simply claims to be the state of the art. The reasons why this paper is NOT sufficient for publication in Nature Communications are described below.

1. IMPACT on METHOD: Despite many papers have been published on thermal cloaking (and its applications) designed by numerical approaches including topology optimization, but there are almost no descriptions of them. The authors should comprehensively cite and organize previous studies using the density method, level set method, and homogenization method (including papers using dehomogenization), and explain the location and advantages of this research. However, no descriptions and paragraph on the previous works are written in the introduction.

We thank the reviewer for the comments. We would like to clarify that in the original draft, we have cited and discussed in detail three papers [42, 43, 26] that use density-based topology optimization for designing thermal metamaterials in the Introduction. To address the concern, we have added comments and citations on level-set topology optimization approaches to thermal metamaterials. We have also included an additional paper using density-based topology optimization and discussed its features in the third paragraph of the Introduction.

As discussed in the Introduction of our manuscript, our study fundamentally departs from the state-of-the-art of both the non-omnidirectional (elaborated in this response comment) and omnidirectional thermal metamaterial (elaborated in the response comment #2 below). We would like to clarify that most topology optimization-based meta-devices belong to a fundamentally different non-omnidirectional class of meta-devices with load case- and direction-sensitive performance. That class of non-omnidirectional meta-devices does not use transformation thermotics, and performance significantly deteriorates when the direction of the applied heat is changed. That class of metamaterial is not the focus of this work and hence is not the main focus of our Introduction discussion (although also cited). We have clarified this point in the added paragraph.

“In addition to transformation-based theories, direct use of topology optimization [34, 35, 36] and data-driven methods for designing global structures [37, 38, 39, 40] or local microstructures [41] can produce the desired functionalities for specific targeted load cases and directions and are thus non-omnidirectional. We note that this class of non-omnidirectional metamaterials does not employ transformation thermotics, and the meta-device’s performance significantly deteriorates when the direction of the applied heat is changed. The scope of this study is the omnidirectional class of thermal metamaterial, where furthermore the conductivity distribution is obtained analytically and free of iterative procedures at a fraction of the cost of above-mentioned numerical approaches.”

“If not solvable by simple intuition, the state-of-the-art strategies to address the de-homogenization challenge is to divide the irregular domains into many square unit cells and use computational morphogenesis approaches (such as topology optimization) with numerical homogenization to inversely design the locally graded unit cell microstructures [42, 43, 26, 44].”

2. Also, this study is very similar to the previous study [24], and the introduction need to include how this study is innovative compared to the study ref. [24] that used topology optimization using the density method for microstructure design. Of course, there are some differences from the references. However, when asked whether this has enough impact to be published in Nature Communications, this reviewer thinks it is NOT.

This study is *fundamentally different from, and much advantageous over [24]* (new reference ordering is [42]) in the following aspects: the local microstructure generation, global structure realization, structure connectivity, and thermal function performance, as evidenced by the following:

- Straightforward analytical realization v.s. computational inversely designed microstructures: The rank-2 microstructures in our structures are obtained analytically with closed-form solutions. There is no design aspect involved in the entire process. The structures in [24] are obtained by topology optimization inverse design involving numerous heavy numerical solutions of PDEs and iterative optimization updates.
- Optimal v.s. sub-optimal performance: In our study, the closed-form solution of rank-2 laminates ensures that any given (physically achievable) target conductivity is achieved *exactly*, thereby ensuring the close-to-optimal performance. By contrast, the microstructures generated by topology optimization in [24] are sub-optimal and may not exactly achieve a given target due to the severe non-convexity of the inverse design problem as well as connectivity issues between cells.
- Guaranteed and post-processing-free structural connectivity v.s. disconnected and post-processing-requiring structures: Our approach of using a pseudo-conformal mapping to generate the global structure naturally ensures connectivity among the members and is free of post-processing, whereas the global structure connectivity in [24] cannot be satisfied without heavy post-processing that in turn perturbs the local homogenized properties.
- Negligible v.s. extremely high computational cost: The main computational cost of our approach for a high-resolution structure is *less than 30 seconds* on a desktop workstation, whereas the approach in [24] requires orders-of-magnitude higher computation cost due to the need for solving many inverse design problems, each of which in turn requires several-hundred numerical solves of the PDE.
- Elegant and aesthetically concise v.s. hard-to-interpret and overly complex patterns: Our global structural patterns (with close-to-optimal performance) are elegant as the member orientations are aligned with the principal directions of the theoretical conductivity and member size proportional to the magnitude of eigenvalues of the conductivity. Further, our global structure is aesthetically concise because each local microstructure (rank-2 laminates) contains only two members. By contrast, the global structures in [24] have many more complex structural branches with highly irregular sizes, shapes, and orientations, which prohibit direct perception of the magnitude and principal directions of the conductivity and complicates the fabrication.
- On a higher and more general level, our study demonstrates an inspirational idea: complex and anisotropic material property distributions can be analytically and optimally realized by extremely simple graded structures. By contrast, study [24] appears to have demonstrated the reverse: realization of complex material properties requires highly sophisticated

techniques and microstructures. The new inspiration from our study is translatable and impactful to many other engineering disciplines.

The above facts demonstrate that the study is fundamentally different from [24] and is not “very similar to the previous study” as suggested in the comment. All the above arguments were already included in our original manuscript, however, in light of the reviewer’s comments, we have carefully gone through the manuscript and edited and sharpened, where appropriate.

3. PERFORMANCE: Despite the very high degree of design freedom by designing the microstructure, the RTD of the designed structure is higher (worsen) than the equivalent value in some previous studies and it has not been improved.

In the state-of-the-art paper [Y. Wang, W. Sha, M. Xiao, C. Qiu, L. Gao. Adv. Mater. 2023, 2302387] on free-form omnidirectional thermal metamaterials, the Relative Temperature Difference (RTD) values of irregular-shape cloaks are higher than 13 % (the regular circular shape has 2.02%). By contrast, using the same RTD definition and with a similar level of shape complexity and irregularity, our corresponding omnidirectional cloaks achieve RTD values of 1.37% and 1.27% (as demonstrated in the figure below), which are much smaller than the current state-of-the-art irregular cloaks in [Y. Wang, W. Sha, M. Xiao, C. Qiu, L. Gao. Adv. Mater. 2023, 2302387]. We note that 0% RTD is the theoretically ideal situation, which cannot be attained by physical materials. By comparing the RTD values, we demonstrate a significant improvement over the state-of-the-art work on omnidirectional thermal metamaterials, and are not “worse than other papers” as claimed in Reviewer 2’s comment about the RTD values.

We would like to clarify that the scope of this paper is the transformation-based, omnidirectional class of thermal metamaterial with performance insensitive to applied heat direction and material of the core (core can be conductive or insulating). Thus, the above comparison to the state-of-the-art in the omnidirectional class is valid. Our scope is not the direction-sensitive class used in many numerical works. For that class of metamaterial, the performance can be very good, if not perfect ($= 0$), exactly in the targeted direction and with an insulating core. However, change of heat direction or core material significantly deteriorates the performance to “inferior” as described in [37], with RTD measures increased to 15% - 80% [37]. Therefore, we do not (and should not) compare our performance with that fundamentally different and simpler class of meta-devices.

To avoid confusion about the metadvice class, we have added the description below to the first paragraph of the Introduction.

“In addition to transformation-based theories, direct use of topology optimization [34, 35, 36] and data-driven methods for designing global structures [37, 38, 39, 40] or local microstructures [41] can produce the desired functionalities for specific targeted load cases

and directions and are thus non-omnidirectional. We note that this class of non-omnidirectional metamaterials does not employ transformation thermotics, and the meta-device's performance significantly deteriorates when the direction of the applied heat is changed. The scope of this study is the omnidirectional class of thermal metamaterial, where furthermore the conductivity distribution is obtained analytically and free of iterative procedures at a fraction of the cost of above-mentioned numerical approaches."

4. QUALITY: (??) remains just before the SI equation (15) and this shows the quality of the manuscript.

We apologize for this editing error. We have corrected it and thoroughly rechecked the revised manuscript.

Reviewer #3 (Remarks to the Author):

The so called "transformation physics" leads to designs, using composites at the microscale, that lead to sometimes surprising conclusions about what is possible. However, the designs are typically far from optimal and sometimes utilize constituent materials with extreme properties. By contrast the authors show how one can use relatively simple designs, incorporating rank 2 laminates at the microscale, that have a continuous morphing of structure on the mesoscale, and they verify their near optimal performance experimentally. This is a major advance and, although currently limited to two-dimensions, there is no reason that the work cannot be extended to three dimensions. However, a major flaw in the paper is that it perpetuates misconceptions about original discoveries.

The authors are most grateful for the reviewers' insightful comments and suggestions and praise of our work that were not mentioned in the common literature. We have thoroughly revised the manuscript according to all the comments. The textual changes are highlighted in red color in the revised manuscript. Below we address each comment in detail.

1. Notably, Pendry, Schurig and Smith were not the first to discover "transformation electromagnetism: this dates back to Dolin ("To the possibility of comparison of three-dimensional electromagnetic systems with nonuniform anisotropic filling", *Izvestiya Vysshikh Uchebnykh Zavedeni\{u\{i\}\} Radiofizika* 4, 964--967 (1961)) who showed how to construct invisible inclusions via this method. A translation of Dolin's paper can be found on the internet. Nor were Pendry, Schurig and Smith and Leonhardt the first to discover cloaking. Transformation conductivity (including thermal conductivity), was used by Greenleaf, Lassas, and Uhlmann in 2003 to cloak conducting objects: (<http://iopscience.iop.org/0967-3334/24/2/353> and <http://dx.doi.org/10.4310/MRL.2003.v10.n5.a11>) In fact the results of Pendry, Schurig, and Smith are a simple combination of these ideas of Dolin and Greenleaf, Lassas, and Uhlmann. For acoustic cloaking it would be important to reference the work of Norris (<http://dx.doi.org/10.1098/rspa.2008.0076>).

We thank the reviewer for pointing out this issue and suggesting the important and original works. We have thoroughly revised the first paragraph to correctly reflect the original works as follows.

"Metamaterials exhibit extreme properties not found in nature and enable exotic functionalities once conceivable only in fiction, such as optically cloaking an object from its environment [1, 2, 3]. Originating in theoretical electromagnetism [1], the idea of transformation-based omnidirectional field manipulation through strategically designed material distribution quickly gained enormous attention and spread to multiple disciplines. Most prominently, cloaking of physical fields [4, 5] are theoretically investigated and experimentally reproduced in the discipline of electromagnetism [4, 2, 3, 6, 7, 8, 9] elasticity [10, 11, 12], acoustics [13, 14, 15, 16, 17], and thermotics [18, 19, 20, 21, 22]. In the latter, exotic functions such as heat cloaking [23, 24, 25, 26, 27], rotating [28], concentrating [29], camouflaging [30, 31], illusion [32], and encrypting [33] have been designed and reproduced experimentally. In addition to transformation-based theories, direct use of topology optimization [34, 35, 36] and data-driven methods for designing global structures [37, 38, 39, 40] or local microstructures [41] can produce the desired functionalities for specific targeted load cases and directions and are thus non-omnidirectional. We note that this class of non-omnidirectional metamaterials does not employ transformation thermotics, and the meta-device's performance significantly deteriorates when the direction of the applied heat is changed. The scope of this study is the

omnidirectional class of thermal metamaterial, where furthermore the conductivity distribution is obtained analytically and free of iterative procedures at a fraction of the cost of above-mentioned numerical approaches.”

2. On the second page it is stated that "the complete spectrum of composite's conductivity can indeed be fully and analytically achieved by simple rank-2 laminates". Nowhere here is stated the all important fact that this is limited to two-dimensional composites (and that rank-3 laminates are needed in three dimensions). The references here should include those of Tartar and Murat (which can be found in books on the theory of composites). Otherwise it is a bit unfair, as Professor Kohn, on a trip to the Soviet Union, had communicated the results of Tartar and Murat to Lurie and Cherkhaev who incorporated them in their work, without attribution (they claim they had proved the bounds in an alternative way but utilized the approach of Murat and Tartar as it was simpler). Lurie and Cherkhaev had independently obtained the two dimensional bounds.

We thank the reviewer for alluding us to this historical insight. We have revised the statement to be rigorous and cited the suggested works [48, 49] as well an additional [50] as shown below.

“Contrary to the indication in [42, 43] that the anisotropic and heterogeneous conductivities are difficult to realize by layered structures, the complete spectrum of a two-dimensional (2D) composite's conductivity can indeed be fully and analytically achieved by simple rank-2 laminates [46, 47, 48, 49, 50].”

3. The authors talk about cloaking and concentrating fields between a hot plate and a cold plate. For the case where the hot and cold plates are fixed, this was solved using rank one laminates by Gibiansky, Lurie, and Cherkhaev and Gibiansky in the paper Zhurnal tekhnicheskoi fiziki 58 (1988) 67–74, english translation in Sov. Phys. Tech. Phys. 33:38–42 (1988). This can be made more clear if one considers the periodic extension of their solution. So that work needs to be cited.

We thank the reviewer's suggestion. We have added the citation [56] (the original version) to the result section Sec. 2.2 of the revised manuscript as follows.

“Higher concentration rates require more extreme and anisotropic κ values as manifested by the larger radial members and smaller circulating members. As a side note, related problems for uni-directional concentration have been solved and require only rank-1 laminates for their realization [56].”

4. A major deficiency is that I do not see the equations of thermal conduction used by the authors stated anywhere. Presumably the authors are not incorporating Joule heating which is a non-linear term, but without some statement of the equations it is hard to know what they are including.

We thank the reviewer for pointing this out. Our physical setup is the 2D thermal conduction problem with temperature-independent thermal conductivity tensor and without heat sources or heat sinks. As suggested by the reviewer, we are not considering Joule heating in the equations.

We have added the equation with the description of the setup in the first paragraph of Results as follows.

“This study focuses on steady-state heat conduction in a 2D medium without heat sources or sinks, which is mathematically described by $\nabla \cdot (\boldsymbol{\kappa} \nabla T) = 0$ with T being the temperature field and $\boldsymbol{\kappa}$ the anisotropic 2x2 temperature-independent thermal conductivity tensor of the medium.”

Additionally, we have added a section (SI 6) in the Supplementary Information about the thermal conduction boundary value problem and the finite element solution setup as follows.

“After obtaining the design of the metamaterial structure, we post-evaluate its heat conduction response using FEM. The steady-state 2D heat conduction problem is

$$\begin{aligned} \nabla \cdot (\boldsymbol{\kappa} \nabla T) &= 0 \quad \text{in } \Omega \\ T &= \bar{T}_H \quad \text{on } \partial\Omega_{T,H} \\ T &= \bar{T}_L \quad \text{on } \partial\Omega_{T,L} \\ -\boldsymbol{\kappa} \nabla T &= 0 \quad \text{on } \partial\Omega_q \end{aligned}$$

where Ω is the domain, $\partial\Omega_{T,H}$ and $\partial\Omega_{T,L}$ are the Dirichlet boundaries of the hot and cold ends, respectively, and $\partial\Omega_q$ are the adiabatic Neumann boundaries, such that $(\partial\Omega_{T,H} \cup \partial\Omega_{T,L}) \cup \partial\Omega_q = \partial\Omega$ and $(\partial\Omega_{T,H} \cup \partial\Omega_{T,L}) \cap \partial\Omega_q = \emptyset$, \bar{T}_H and \bar{T}_L are the prescribed temperature for the hot and cold ends, respectively. Note that we focus on the case with no internal heat sources or sinks.

The boundary value problem is solved using an in-house FEM program where the four-node quadrilateral element with four Gauss points is adopted. The $\boldsymbol{\kappa}$ is assumed to be piece-wise constant and associated with the elements. Specifically, $\boldsymbol{\kappa} = \kappa_{\text{Steel}} \mathbf{I}$ for the steel parts of the structure, $\boldsymbol{\kappa} = \kappa_{\text{PDMS}} \mathbf{I}$ for the PDMS parts, and $\boldsymbol{\kappa} = \kappa_0 \mathbf{I}$ for the background. The square-shaped domain is discretized by a regular 2000x2000 mesh to allow for sufficient resolution of the microstructure.”

General comment:

In the revision process, we have carefully reread the entire manuscript and made changes throughout that we believe have sharpened our messages.

References

- [1] L. Dolin. To the possibility of comparison of three-dimensional electromagnetic systems with nonuniform anisotropic filling. *Izv. Vyssh. Uchebn. Zaved. Radiofiz.*, 4:964–967, 01 1961.
- [2] J. B. Pendry, D. Schurig, and D. R. Smith. Controlling electromagnetic fields. *Science*, 312(5781):1780–1782, 2006.
- [3] Ulf Leonhardt. Optical conformal mapping. *Science*, 312(5781):1777–1780, 2006.
- [4] Allan Greenleaf, Matti Lassas, and Gunther Uhlmann. Anisotropic conductivities that cannot be detected by eit. *Physiological Measurement*, 24(2):413, apr 2003.
- [5] Allan Greenleaf, Matti Lassas, and Gunther Uhlmann. On nonuniqueness for calderón’s inverse problem. *Mathematical Research Letters*, 10:685–693, 2003.
- [6] D. Schurig, J. J. Mock, B. J. Justice, S. A. Cummer, J. B. Pendry, A. F. Starr, and D. R. Smith. Metamaterial electromagnetic cloak at microwave frequencies. *Science*, 314(5801):977–980, 2006.
- [7] Jason Valentine, Jensen Li, Thomas Zentgraf, Guy Bartal, and Xiang Zhang. An optical cloak made of dielectrics. *Nature Materials*, 8(7):568–571, Jul 2009.
- [8] Tolga Ergin, Nicolas Stenger, Patrice Brenner, John B. Pendry, and Martin Wegener. Three-dimensional invisibility cloak at optical wavelengths. *Science*, 328(5976):337–339, 2010.
- [9] Fan Yang, Zhong Lei Mei, Tian Yu Jin, and Tie Jun Cui. dc electric invisibility cloak. *Phys. Rev. Lett.*, 109:053902, Aug 2012.
- [10] Nicolas Stenger, Manfred Wilhelm, and Martin Wegener. Experiments on elastic cloaking in thin plates. *Phys. Rev. Lett.*, 108:014301, Jan 2012.
- [11] T. Bückmann, M. Thiel, M. Kadic, R. Schittny, and M. Wegener. An elasto-mechanical unfeelability cloak made of pentamode metamaterials. *Nature Communications*, 5(1):4130, Jun 2014.
- [12] Liwei Wang, Jagannadh Boddapati, Ke Liu, Ping Zhu, Chiara Daraio, and Wei Chen. Mechanical cloak via data-driven aperiodic metamaterial design. *Proceedings of the National Academy of Sciences*, 119(13):e2122185119, 2022.
- [13] Huanyang Chen and C. T. Chan. Acoustic cloaking in three dimensions using acoustic metamaterials. *Applied Physics Letters*, 91(18):183518, 11 2007.
- [14] Andrew N Norris. Acoustic cloaking theory. *Proceedings of the Royal Society A: Mathematical, Physical and Engineering Sciences*, 464(2097):2411–2434, 2008.
- [15] M. Farhat, S. Enoch, S. Guenneau, and A. B. Movchan. Broadband cylindrical acoustic cloak for linear surface waves in a fluid. *Phys. Rev. Lett.*, 101:134501, Sep 2008.
- [16] Bogdan-Ioan Popa, Lucian Zigoneanu, and Steven A. Cummer. Experimental acoustic ground cloak in air. *Phys. Rev. Lett.*, 106:253901, Jun 2011.
- [17] Shu Zhang, Chunguang Xia, and Nicholas Fang. Broadband acoustic cloak for ultrasound waves. *Phys. Rev. Lett.*, 106:024301, Jan 2011.
- [18] C. Z. Fan, Y. Gao, and J. P. Huang. Shaped graded materials with an apparent negative thermal conductivity. *Applied Physics Letters*, 92(25):251907, 06 2008.
- [19] Xiangying Shen, Ying Li, Chaoran Jiang, and Jiping Huang. Temperature trapping: Energy-free maintenance of constant temperatures as ambient temperature gradients change. *Phys. Rev. Lett.*, 117:055501, Jul 2016.
- [20] Ji-Ping Huang. *Theoretical Thermotics: Transformation Thermotics and Extended Theories for Thermal Metamaterials*. Springer, 2020.

- [21] Run Hu, Wang Xi, Yida Liu, Kechao Tang, Jinlin Song, Xiaobing Luo, Junqiao Wu, and Cheng-Wei Qiu. Thermal camouflaging metamaterials. *Materials Today*, 45:120–141, 2021.
- [22] Min Lei, Chaoran Jiang, Fubao Yang, Jun Wang, and Jiping Huang. Programmable all-thermal encoding with metamaterials. *International Journal of Heat and Mass Transfer*, 207:124033, 2023.
- [23] Tiancheng Han, Xue Bai, Dongliang Gao, John T. L. Thong, Baowen Li, and Cheng-Wei Qiu. Experimental demonstration of a bilayer thermal cloak. *Phys. Rev. Lett.*, 112:054302, Feb 2014.
- [24] Tiancheng Han, Peng Yang, Ying Li, Dangyuan Lei, Baowen Li, Kedar Hippalgaonkar, and Cheng-Wei Qiu. Full-parameter omnidirectional thermal metadevices of anisotropic geometry. *Advanced Materials*, 30(49):1804019, 2018.
- [25] Zhan Zhu, Xuecheng Ren, Wei Sha, Mi Xiao, Run Hu, and Xiaobing Luo. Inverse design of rotating metadvice for adaptive thermal cloaking. *International Journal of Heat and Mass Transfer*, 176:121417, 2021.
- [26] Zhan Zhu, Zhaochen Wang, Tianfeng Liu, Xiaobing Luo, Chengwei Qiu, and Run Hu. Field-coupling topology design of general transformation multiphysics metamaterials with different functions and arbitrary shapes. *Cell Reports Physical Science*, 4(8):101540, 2023.
- [27] Kazuma Hirasawa, Iona Nakami, Takumi Ooinoue, Tatsunori Asaoka, and Garuda Fujii. Experimental demonstration of thermal cloaking metastructures designed by topology optimization. *International Journal of Heat and Mass Transfer*, 194:123093, 2022.
- [28] Fubao Yang, Boyan Tian, Liujun Xu, and Jiping Huang. Experimental demonstration of thermal chameleonlike rotators with transformation-invariant metamaterials. *Phys. Rev. Appl.*, 14:054024, Nov 2020.
- [29] Xiangying Shen, Ying Li, Chaoran Jiang, Yushan Ni, and Jiping Huang. Thermal cloak-concentrator. *Applied Physics Letters*, 109(3):031907, 07 2016.
- [30] Yurui Qu, Qiang Li, Lu Cai, Meiyang Pan, Pintu Ghosh, Kaikai Du, and Min Qiu. Thermal camouflage based on the phase-changing material gst. *Light: Science & Applications*, 7(1):26, Jun 2018.
- [31] Sahngki Hong, Sunmi Shin, and Renkun Chen. An adaptive and wearable thermal camouflage device. *Advanced Functional Materials*, 30(11):1909788, 2020.
- [32] Run Hu, Shuling Zhou, Ying Li, Dang-Yuan Lei, Xiaobing Luo, and Cheng-Wei Qiu. Illusion thermotics. *Advanced Materials*, 30(22):1707237, 2018.
- [33] Run Hu, Shiyao Huang, Meng Wang, Xiaobing Luo, Junichiro Shiomi, and Cheng-Wei Qiu. Encrypted thermal printing with regionalization transformation. *Advanced Materials*, 31(25):1807849, 2019.
- [34] Martin Philip Bendsøe and Noboru Kikuchi. Generating optimal topologies in structural design using a homogenization method. *Computer Methods in Applied Mechanics and Engineering*, 71(2):197 – 224, 1988.
- [35] Martin P. Bendsøe and Ole Sigmund. *Topology Optimization: Theory, Methods and Applications*. Springer, Berlin, Heidelberg, 2003.
- [36] Michael Yu Wang, Xiaoming Wang, and Dongming Guo. A level set method for structural topology optimization. *Computer Methods in Applied Mechanics and Engineering*, 192(1):227–246, 2003.
- [37] Garuda Fujii, Youhei Akimoto, and Masayuki Takahashi. Exploring optimal topology of thermal cloaks by CMA-ES. *Applied Physics Letters*, 112(6):061108, 02 2018.
- [38] Garuda Fujii and Youhei Akimoto. Cloaking a concentrator in thermal conduction via topology optimization. *International Journal of Heat and Mass Transfer*, 159:120082, 2020.

- [39] Garuda Fujii. Biphysical undetectable concentrators manipulating both heat flux and direct current via topology optimization. *Phys. Rev. E*, 106:065304, Dec 2022.
- [40] Ji-Wang Luo, Li Chen, Zihan Wang, and WenQuan Tao. Topology optimization of thermal cloak using the adjoint lattice boltzmann method and the level-set method. *Applied Thermal Engineering*, 216:119103, 2022.
- [41] Daicong Da and Wei Chen. Two-scale data-driven design for heat manipulation. *International Journal of Heat and Mass Transfer*, 219:124823, 2024.
- [42] Wei Sha, Mi Xiao, Jinhao Zhang, Xuecheng Ren, Zhan Zhu, Yan Zhang, Guoqiang Xu, Huagen Li, Xiliang Liu, Xia Chen, Liang Gao, Cheng-Wei Qiu, and Run Hu. Robustly printable freeform thermal metamaterials. *Nature Communications*, 12(1):7228, Dec 2021.
- [43] Wei Sha, Mi Xiao, Mingzhe Huang, and Liang Gao. Topology-optimized freeform thermal meta- materials for omnidirectionally cloaking sensors. *Materials Today Physics*, 28:100880, 2022.
- [44] Zhan Zhu, Zhaochen Wang, Tianfeng Liu, Bin Xie, Xiaobing Luo, Wonjoon Choi, and Run Hu. Arbitrary-shape transformation multiphysics cloak by topology optimization. *International Journal of Heat and Mass Transfer*, 222:125205, 2024.
- [45] Yihui Wang, Wei Sha, Mi Xiao, Cheng-Wei Qiu, and Liang Gao. Deep-learning-enabled intelligent design of thermal metamaterials. *Advanced Materials*, 35(33):2302387, 2023.
- [46] K. A. Lurie and A. V. Cherkhaev. Exact estimates of conductivity of composites formed by two isotropically conducting media taken in prescribed proportion. *Proceedings of the Royal Society of Edinburgh: Section A Mathematics*, 99(1–2):71–87, 1984.
- [47] K. A. Lurie and A. V. Cherkhaev. Exact estimates of the conductivity of a binary mixture of isotropic materials. *Proceedings of the Royal Society of Edinburgh: Section A Mathematics*, 104(1–2):21–38, 1986.
- [48] Francois Murat and Luc Tartar. *H-Convergence*, pages 21–43. Birkhäuser Boston, Boston, MA, 1997.
- [49] Luc Tartar. *Estimations of Homogenized Coefficients*, pages 9–20. Birkhäuser Boston, Boston, MA, 1997.
- [50] Andrej Cherkhaev. *Variational Methods for Structural Optimization*. Springer-Verlag, Berlin / Heidelberg / New York / etc., 2000.
- [51] O. Pantz and K. Trabelsi. A post-treatment of the homogenization method for shape optimization. *SIAM Journal on Control and Optimization*, 47(3):1380–1398, 2008.
- [52] Jeroen P. Groen and Ole Sigmund. Homogenization-based topology optimization for high- resolution manufacturable microstructures. *International Journal for Numerical Methods in Engineering*, 113(8):1148–1163, 2018.
- [53] Grégoire Allaire, Perle Geoffroy-Donders, and Olivier Pantz. Topology optimization of modulated and oriented periodic microstructures by the homogenization method. *Computers Mathematics with Applications*, 78(7):2197–2229, 2019. Simulation for Additive Manufacturing.
- [54] Jeroen P. Groen, Florian C. Stutz, Niels Aage, Jakob A. Bærentzen, and Ole Sigmund. De- homogenization of optimal multi-scale 3d topologies. *Computer Methods in Applied Mechanics and Engineering*, 364:112979, 2020.
- [55] Peter Dørffler Ladegaard Jensen, Ole Sigmund, and Jeroen P. Groen. De-homogenization of optimal 2d topologies for multiple loading cases. *Computer Methods in Applied Mechanics and Engineering*, 399:115426, 2022.
- [56] L.V. Gibyanskii, K.A. Lure, and A.V. Cherkhaev. Optimum focusing of heat flux by means of a non-homogeneous heat-conducting medium. *Zhurnal Tekhnicheskoi Fiziki*, 58(1):67–74, 1988.

REVIEWERS' COMMENTS

Reviewer #1 (Remarks to the Author):

The authors have well addressed my previous concerns and it is recommended for publication now.

Reviewer #2 (Remarks to the Author):

I read the Response and reconsider the importance of the presented originality in the manuscript.

Major Comments

1. IMPACT & PERFORMANCE: Indeed, as stated in authors' reply to my comment 2, the method presented in this manuscript has several advantages compared to the Ref. [42]. On the other hand, if my understand is correct, all the above advantages is achieved by sacrificing the degree of freedom in designing the global structure described by wave function representation in Eq. (2). Although it is true that Ref. [42] has a high computational cost and may not have sufficient performance compared to the results of this manuscript, it also has an extremely high degree of freedom in design, which is one of the original academic appeals of metamaterials.

The main premise is that a omni directional thermal cloak that can make object undetectable from heat flux for all directions can be realized with a very simple structure like a bi-layer cloak in Ref. [23], so it is not difficult to realize its function. Of course, by normalizing the equation, its performance is also independent of the magnitude of the heat flux.

In that sense, the proposed method is inferior to Ref. [42] in terms of design freedom, and inferior to bilayer cloaks in terms of structural simplicity (ease of fabrication) and performance. For numerical performance evaluation of the bilayer cloak, see the end of Ref. [37].

The method presented in this manuscript achieves the above advantages by sacrificing the design freedom, but are they worthy of publication in Nature Communications?

Minor Comments

2. What is ρ used to express the geometry in Eq. (2) ? According to Ref. [53], ρ seems to be a level set function, but is this not the case? There might be no definition.

3. Since we are talking about heat conduction here, the expression "load" is confusing.

4. The related papers on thermal cloaking should be included in introduction as H. Xu et al. Phys. Rev. Lett. 112, 054301 (2014); G. Fujii et al. Int. J. Heat Mass Transf. 137, 1312-1322 (2019); M. Nakagawa et al., Int. J. Heat Mass Transf. 207, 123964 (2023).

Reviewer #3 (Remarks to the Author):

The authors have satisfactorily responded to my comments, and I recommend publication in Nature.

Response to the reviewers

The authors are grateful for the reviewers' recognition and additional insightful comments. We have addressed the comments and revised accordingly. The textual changes are highlighted in red color in the revised manuscript and the added citations are highlighted yellow. Below we address each comment in detail.

Reviewer #2 (Remarks to the Author):

I read the Response and reconsider the importance of the presented originality in the manuscript.

Major Comments:

1. IMPACT & PERFORMANCE: Indeed, as stated in authors' reply to my comment 2, the method presented in this manuscript has several advantages compared to the Ref. [42]. On the other hand, if my understand is correct, all the above advantages is achieved by sacrificing the degree of freedom in designing the global structure described by wave function representation in Eq. (2). Although it is true that Ref. [42] has a high computational cost and may not have sufficient performance compared to the results of this manuscript, it also has an extremely high degree of freedom in design, which is one of the original academic appeals of metamaterials.

The main premise is that a omni directional thermal cloak that can make object undetectable from heat flux for all directions can be realized with a very simple structure like a bi-layer cloak in Ref. [23], so it is not difficult to realize its function. Of course, by normalizing the equation, its performance is also independent of the magnitude of the heat flux.

In that sense, the proposed method is inferior to Ref. [42] in terms of design freedom, and inferior to bilayer cloaks in terms of structural simplicity (ease of fabrication) and performance. For numerical performance evaluation of the bilayer cloak, see the end of Ref. [37].

The method presented in this manuscript achieves the above advantages by sacrificing the design freedom, but are they worthy of publication in Nature Communications?

We thank the reviewer for raising these important points. We would like to clarify that this work did not sacrifice any design freedom, because the rank-2 laminate has the full design freedom in that it achieves all physically possible thermal conductivities for a 2D composite, as indicated in SI Figure 2 (attached below). This has been theoretically proved [48 - 52]. The geometric simplicity of rank-2 laminate is an advantage (rather than a disadvantage) over the complex microstructures, and it is not "*inferior to Ref. [42] in terms of design freedom*" nor has it "*sacrificed design freedom*".

To clarify this point, we have modified the comment on rank-2 laminates in the second paragraph of the Results section as follows:

"With their simple microstructures, rank-2 laminates have been shown to cover the entire range of physically achievable thermal conductivities for 2D composites [52], and therefore, our approach does not sacrifice any design freedom."

For the bi-layer cloak in [23], we are aware that it can achieve cloaking but only for very simple geometry, such as the circular/spherical domain and not for general domains. The scope and focus of this study have always been cloaks, rotators, and concentrators with general and complex geometries, which has been a major challenge in the field because of the required highly anisotropic and graded conductivities as mentioned in recent studies such as [26, 44,

45]. Cloaks of general and complex geometries are thus not “*not difficult to realize*”, and they are effectively and innovatively achieved in this study. Our study tackles a fundamentally different and more general scenario that cannot be realized by the bi-layer scheme [23].

Minor Comments

2. What is ρ used to express the geometry in Eq. (2)? According to Ref. [53], ρ seems to be a level set function, but is this not the case? There might be no definition.

We thank the reviewer for pointing at this issue. It is indeed a level set function and was termed “wave function” in our study. We have modified the term to “wave level set function” throughout to be more consistent with [55] (ref. [53] in the original draft).

3. Since we are talking about heat conduction here, the expression “load” is confusing.

We thank the reviewer for pointing this out. We have changed “load” to “boundary conditions” as shown below (in the Introduction).

“... can produce the desired functionalities for specific targeted boundary conditions and applied heat gradient directions and are thus non-omnidirectional.”

4. The related papers on thermal cloaking should be included in introduction as H. Xu et al. Phys. Rev. Lett. 112, 054301 (2014); G. Fujii et al. Int. J. Heat Mass Transf. 137, 1312-1322 (2019); M. Nakagawa et al., Int. J. Heat Mass Transf. 207, 123964 (2023).

We thank the reviewer’s suggestion. The paper H. Xu et al. Phys. Rev. Lett. 112, 054301 (2014) has already been cited in our original draft, and the latter two are cited in the revised manuscript as follows (highlighted).

“In addition to transformation-based theories, direct use of topology optimization [34, 35, 36] and data-driven methods for generating global structures [37, 38, 39, 40, 41] or local microstructures [42, 43] produce the desired functionalities for specific targeted boundary conditions and applied heat directions and are thus non-omnidirectional.”

References

- [1] L. Dolin. To the possibility of comparison of three-dimensional electromagnetic systems with nonuniform anisotropic filling. *Izv. Vyssh. Uchebn. Zaved. Radiofiz.*, 4:964–967, 01 1961.
- [2] J. B. Pendry, D. Schurig, and D. R. Smith. Controlling electromagnetic fields. *Science*, 312(5781):1780–1782, 2006.
- [3] Ulf Leonhardt. Optical conformal mapping. *Science*, 312(5781):1777–1780, 2006.
- [4] Allan Greenleaf, Matti Lassas, and Gunther Uhlmann. Anisotropic conductivities that cannot be detected by eit. *Physiological Measurement*, 24(2):413, apr 2003.
- [5] Allan Greenleaf, Matti Lassas, and Gunther Uhlmann. On nonuniqueness for calderón’s inverse problem. *Mathematical Research Letters*, 10:685–693, 2003.
- [6] D. Schurig, J. J. Mock, B. J. Justice, S. A. Cummer, J. B. Pendry, A. F. Starr, and D. R. Smith. Metamaterial electromagnetic cloak at microwave frequencies. *Science*, 314(5801):977–980, 2006.
- [7] Jason Valentine, Jensen Li, Thomas Zentgraf, Guy Bartal, and Xiang Zhang. An optical cloak made of dielectrics. *Nature Materials*, 8(7):568–571, Jul 2009.
- [8] Tolga Ergin, Nicolas Stenger, Patrice Brenner, John B. Pendry, and Martin Wegener. Three-dimensional invisibility cloak at optical wavelengths. *Science*, 328(5976):337–339, 2010.
- [9] Fan Yang, Zhong Lei Mei, Tian Yu Jin, and Tie Jun Cui. dc electric invisibility cloak. *Phys. Rev. Lett.*, 109:053902, Aug 2012.
- [10] Nicolas Stenger, Manfred Wilhelm, and Martin Wegener. Experiments on elastic cloaking in thin plates. *Phys. Rev. Lett.*, 108:014301, Jan 2012.
- [11] T. Bückmann, M. Thiel, M. Kadic, R. Schittny, and M. Wegener. An elasto-mechanical unfeelability cloak made of pentamode metamaterials. *Nature Communications*, 5(1):4130, Jun 2014.
- [12] Liwei Wang, Jagannadh Boddapati, Ke Liu, Ping Zhu, Chiara Daraio, and Wei Chen. Mechanical cloak via data-driven aperiodic metamaterial design. *Proceedings of the National Academy of Sciences*, 119(13):e2122185119, 2022.
- [13] Huanyang Chen and C. T. Chan. Acoustic cloaking in three dimensions using acoustic metamaterials. *Applied Physics Letters*, 91(18):183518, 11 2007.
- [14] Andrew N Norris. Acoustic cloaking theory. *Proceedings of the Royal Society A: Mathematical, Physical and Engineering Sciences*, 464(2097):2411–2434, 2008.
- [15] M. Farhat, S. Enoch, S. Guenneau, and A. B. Movchan. Broadband cylindrical acoustic cloak for linear surface waves in a fluid. *Phys. Rev. Lett.*, 101:134501, Sep 2008.
- [16] Bogdan-Ioan Popa, Lucian Zigoneanu, and Steven A. Cummer. Experimental acoustic ground cloak in air. *Phys. Rev. Lett.*, 106:253901, Jun 2011.
- [17] Shu Zhang, Chunguang Xia, and Nicholas Fang. Broadband acoustic cloak for ultrasound waves. *Phys. Rev. Lett.*, 106:024301, Jan 2011.
- [18] C. Z. Fan, Y. Gao, and J. P. Huang. Shaped graded materials with an apparent negative thermal conductivity. *Applied Physics Letters*, 92(25):251907, 06 2008.
- [19] Xiangying Shen, Ying Li, Chaoran Jiang, and Jiping Huang. Temperature trapping: Energy-free maintenance of constant temperatures as ambient temperature gradients change. *Phys. Rev. Lett.*, 117:055501, Jul 2016.
- [20] Ji-Ping Huang. *Theoretical Thermotics: Transformation Thermotics and Extended Theories for Thermal Metamaterials*. Springer, 2020.

- [21] Run Hu, Wang Xi, Yida Liu, Kechao Tang, Jinlin Song, Xiaobing Luo, Junqiao Wu, and Cheng-Wei Qiu. Thermal camouflaging metamaterials. *Materials Today*, 45:120–141, 2021.
- [22] Min Lei, Chaoran Jiang, Fubao Yang, Jun Wang, and Jiping Huang. Programmable all-thermal encoding with metamaterials. *International Journal of Heat and Mass Transfer*, 207:124033, 2023.
- [23] Tiancheng Han, Xue Bai, Dongliang Gao, John T. L. Thong, Baowen Li, and Cheng-Wei Qiu. Experimental demonstration of a bilayer thermal cloak. *Phys. Rev. Lett.*, 112:054302, Feb 2014.
- [24] Tiancheng Han, Peng Yang, Ying Li, Dangyuan Lei, Baowen Li, Kedar Hippalgaonkar, and Cheng-Wei Qiu. Full-parameter omnidirectional thermal metadevices of anisotropic geometry. *Advanced Materials*, 30(49):1804019, 2018.
- [25] Zhan Zhu, Xuecheng Ren, Wei Sha, Mi Xiao, Run Hu, and Xiaobing Luo. Inverse design of rotating metadvice for adaptive thermal cloaking. *International Journal of Heat and Mass Transfer*, 176:121417, 2021.
- [26] Zhan Zhu, Zhaochen Wang, Tianfeng Liu, Xiaobing Luo, Chengwei Qiu, and Run Hu. Field-coupling topology design of general transformation multiphysics metamaterials with different functions and arbitrary shapes. *Cell Reports Physical Science*, 4(8):101540, 2023.
- [27] Kazuma Hirasawa, Iona Nakami, Takumi Ooinoue, Tatsunori Asaoka, and Garuda Fujii. Experimental demonstration of thermal cloaking metastructures designed by topology optimization. *International Journal of Heat and Mass Transfer*, 194:123093, 2022.
- [28] Fubao Yang, Boyan Tian, Liuju Xu, and Jiping Huang. Experimental demonstration of thermal chameleonlike rotators with transformation-invariant metamaterials. *Phys. Rev. Appl.*, 14:054024, Nov 2020.
- [29] Xiangying Shen, Ying Li, Chaoran Jiang, Yushan Ni, and Jiping Huang. Thermal cloak-concentrator. *Applied Physics Letters*, 109(3):031907, 07 2016.
- [30] Yurui Qu, Qiang Li, Lu Cai, Meiyang Pan, Pintu Ghosh, Kaikai Du, and Min Qiu. Thermal camouflage based on the phase-changing material gst. *Light: Science & Applications*, 7(1):26, Jun 2018.
- [31] Sahngki Hong, Sunmi Shin, and Renkun Chen. An adaptive and wearable thermal camouflage device. *Advanced Functional Materials*, 30(11):1909788, 2020.
- [32] Run Hu, Shuling Zhou, Ying Li, Dang-Yuan Lei, Xiaobing Luo, and Cheng-Wei Qiu. Illusion thermotics. *Advanced Materials*, 30(22):1707237, 2018.
- [33] Run Hu, Shiyao Huang, Meng Wang, Xiaobing Luo, Junichiro Shiomi, and Cheng-Wei Qiu. Encrypted thermal printing with regionalization transformation. *Advanced Materials*, 31(25):1807849, 2019.
- [34] Martin Philip Bendsøe and Noboru Kikuchi. Generating optimal topologies in structural design using a homogenization method. *Computer Methods in Applied Mechanics and Engineering*, 71(2):197 – 224, 1988.
- [35] Martin P. Bendsøe and Ole Sigmund. *Topology Optimization: Theory, Methods and Applications*. Springer, Berlin, Heidelberg, 2003.
- [36] Michael Yu Wang, Xiaoming Wang, and Dongming Guo. A level set method for structural topology optimization. *Computer Methods in Applied Mechanics and Engineering*, 192(1):227–246, 2003.
- [37] Garuda Fujii, Youhei Akimoto, and Masayuki Takahashi. Exploring optimal topology of thermal cloaks by CMA-ES. *Applied Physics Letters*, 112(6):061108, 02 2018.
- [38] Garuda Fujii and Youhei Akimoto. Topology-optimized thermal carpet cloak expressed by an immersed-boundary level-set method via a covariance matrix adaptation evolution strategy. *International Journal of Heat and Mass Transfer*, 137:1312–1322, 2019.

- [39] Garuda Fujii and Youhei Akimoto. Cloaking a concentrator in thermal conduction via topology optimization. *International Journal of Heat and Mass Transfer*, 159:120082, 2020.
- [40] Garuda Fujii. Biphasical undetectable concentrators manipulating both heat flux and direct current via topology optimization. *Phys. Rev. E*, 106:065304, Dec 2022.
- [41] Ji-Wang Luo, Li Chen, Zihan Wang, and Wenquan Tao. Topology optimization of thermal cloak using the adjoint lattice boltzmann method and the level-set method. *Applied Thermal Engineering*, 216:119103, 2022.
- [42] Makoto Nakagawa, Yuki Noguchi, Kei Matsushima, and Takayuki Yamada. Level set-based multi-scale topology optimization for a thermal cloak design problem using the homogenization method. *International Journal of Heat and Mass Transfer*, 207:123964, 2023.
- [43] Daicong Da and Wei Chen. Two-scale data-driven design for heat manipulation. *International Journal of Heat and Mass Transfer*, 219:124823, 2024.
- [44] Wei Sha, Mi Xiao, Jinhao Zhang, Xuecheng Ren, Zhan Zhu, Yan Zhang, Guoqiang Xu, Huagen Li, Xiliang Liu, Xia Chen, Liang Gao, Cheng-Wei Qiu, and Run Hu. Robustly printable freeform thermal metamaterials. *Nature Communications*, 12(1):7228, Dec 2021.
- [45] Wei Sha, Mi Xiao, Mingzhe Huang, and Liang Gao. Topology-optimized freeform thermal metamaterials for omnidirectionally cloaking sensors. *Materials Today Physics*, 28:100880, 2022.
- [46] Zhan Zhu, Zhaochen Wang, Tianfeng Liu, Bin Xie, Xiaobing Luo, Wonjoon Choi, and Run Hu. Arbitrary-shape transformation multiphysics cloak by topology optimization. *International Journal of Heat and Mass Transfer*, 222:125205, 2024.
- [47] Yihui Wang, Wei Sha, Mi Xiao, Cheng-Wei Qiu, and Liang Gao. Deep-learning-enabled intelligent design of thermal metamaterials. *Advanced Materials*, 35(33):2302387, 2023.
- [48] K. A. Lurie and A. V. Cherkaev. Exact estimates of conductivity of composites formed by two isotropically conducting media taken in prescribed proportion. *Proceedings of the Royal Society of Edinburgh: Section A Mathematics*, 99(1–2):71–87, 1984.
- [49] K. A. Lurie and A. V. Cherkaev. Exact estimates of the conductivity of a binary mixture of isotropic materials. *Proceedings of the Royal Society of Edinburgh: Section A Mathematics*, 104(1–2):21–38, 1986.
- [50] Fran,cois Murat and Luc Tartar. *H-Convergence*, pages 21–43. Birkhäuser Boston, Boston, MA, 1997.
- [51] Luc Tartar. *Estimations of Homogenized Coefficients*, pages 9–20. Birkhäuser Boston, Boston, MA, 1997.
- [52] Andrej Cherkaev. *Variational Methods for Structural Optimization*. Springer-Verlag, Berlin / Heidelberg / New York / etc., 2000.
- [53] O. Pantz and K. Trabelsi. A post-treatment of the homogenization method for shape optimization. *SIAM Journal on Control and Optimization*, 47(3):1380–1398, 2008.
- [54] Jeroen P. Groen and Ole Sigmund. Homogenization-based topology optimization for high-resolution manufacturable microstructures. *International Journal for Numerical Methods in Engineering*, 113(8):1148–1163, 2018.
- [55] Grégoire Allaire, Perle Geoffroy-Donders, and Olivier Pantz. Topology optimization of modulated and oriented periodic microstructures by the homogenization method. *Computers Mathematics with Applications*, 78(7):2197–2229, 2019.
- [56] Jeroen P. Groen, Florian C. Stutz, Niels Aage, Jakob A. Bærentzen, and Ole Sigmund. De-homogenization of optimal multi-scale 3d topologies. *Computer Methods in Applied Mechanics and Engineering*, 364:112979, 2020.

- [57] Peter Dørffler Ladegaard Jensen, Ole Sigmund, and Jeroen P. Groen. De-homogenization of optimal 2d topologies for multiple loading cases. *Computer Methods in Applied Mechanics and Engineering*, 399:115426, 2022.
- [58] L.V. Gibyanskii, K.A. Lurie, and A.V. Cherkaev. Optimum focusing of heat flux by means of a non-homogeneous heat-conducting medium. *Zhurnal Tekhnicheskoi Fiziki*, 58(1):67–74, 1988.